

1   **A coupled soilscape-landform evolution model: Model formulation**

2   **and initial results.**

W. D. Dimuth P.  Welivitiya[1,2], Garry R Willgoose[1], Greg R Hancock[2]
[1]School of Engineering, The University of Newcastle, Callaghan, 2308, Australia.
[2]School of Environment and Life Sciences, The University of Newcastle, Callaghan, 2308,
Australia.
Corresponding Author:       Garry Willgoose

12                               Garry.willgoose@newcastle.edu.au



**Abstract**
This paper describes the coupling of the State Space Soil Production and Assessment Model
(SSSPAM) soilscape evolution model with a landform evolution model to integrate soil profile
dynamics and landform evolution. SSSPAM is a computationally efficient soil evolution model which
was formulated by generalising the mARM3D modelling framework to further explore the soil profile
self-organization in space and time, and its dynamic evolution. The landform evolution was integrated
into SSSPAM by incorporating the processes of deposition and elevation changes resulting from
erosion and deposition. The complexities of the physically based process equations were simplified by
introducing state-space matrix methodology that allows efficient simulation of mechanistically linked
landscape and pedogenesis processes for catena spatial scales. The modelling approach and the
physics underpinning the modelled processes are described in detail. SSSPAM explicitly describes the
particle size grading of the entire soil profile at different soil depths, tracks the sediment grading of
the flow, and calculates the elevation difference caused by erosion and deposition at every point in the
soilscape at each time step. The landform evolution model allows the landform to change in response
to (1) erosion and deposition, and (2) spatial organisation of the co-evolving soils. This allows
comprehensive analysis of soil landform interactions and soil self-organization. SSSPAM simulates
fluvial erosion, armouring, physical weathering, and sediment deposition. The modular nature of the
SSSPAM framework allows integration of other pedogenesis processes in follow-on research projects.
This paper presents the initial results of soil profile evolution on a dynamic landform. These
simulations were carried out on a simple linear hillslope to understand the relationships between soil
characteristics and the geomorphic attributes (e.g. slope, area). Process interactions which lead to such
relationships were also identified. The influence of the depth dependent weathering function on
soilscape and landform evolution was also explored. These simulations show that the balance between
erosion rate and sediment load in the flow accounts for the variability in spatial soil characteristics
while the depth dependent weathering function has a major influence on soil formation and landform
evolution.










**1. Introduction**

Soil is one of the most important substances found on planet Earth. As the uppermost

layer of the earth surface, soil supports all the terrestrial organisms ranging from microbes to
plants to humans and provides the substrate for terrestrial life [*Lin*, 2011]. Soil provides a
transport and a storage medium for water and gases (e.g. carbon dioxide which influence the
global climate) [*Strahler and Strahler*, 2006]. The nature of the soil heavily influences both
geomorphological and hydrological processes [*Bryan*, 2000]. In addition to the importance of
soil from an environmental standpoint, it provides a basis for human civilization and played
an important role in its advancement through the means of agricultural development [*Jenny*,
1941]. Understanding the formation and the global distribution of soil (and its functional
properties) is imperative in the quest for sustainable use of this resource.

Characterization of soil properties at a global scale by sampling and analysis is time

consuming and prohibitively expensive due to the dynamic nature of the soil system and its
complexity [*Hillel*, 1982]. However over the years researchers have found strong links
between different soil properties and between soil properties and geomorphology of the
landform on which they reside [*Gessler et al.* 2000, 1995]. Working on this hypothesis
several statistical methods have been developed to determine and map various soil properties
depending on other soil properties and geomorphology such as pedotransfer functions,
geostatistical approaches, and state-factor (e.g. Clorpt) approaches [*Behrens and Scholten*,
2006]. Pedotransfer functions (PTFs) use easily measurable soil attributes such as particle
size distribution, amount of organic matter, and clay content to predict hard to measure soil
properties such as soil water content. Although very useful, PTFs need a large database of
spatially distributed soil property data and require site specific calibration [*Benites et al.*,
2007]. Geostatistical methods uses a finite number of field samples to interpolate the soil
property distribution over a large area. Developing soil property maps using geostatistical
methods are possible for smaller spatial scales, however soil sampling and mapping soil
attributes can be prohibitively expensive and time consuming for larger spatial domains
[*Scull et al.*, 2003]. State-factor methods, such as Clorpt and Scorpan use digitized existing
soil maps and easily measurable soil attributes data to generate spatially distributed soil
property data using mathematical concepts such as fuzzy set theory, artificial neural network





or decision tree methods [*McBratney et al.*, 2003]. However these techniques also suffer from
scalability issues and the typical need for site specific calibration.

While spatial mapping of soil properties is important, understanding the evolution of

these soil properties and processes responsible for observed spatial variability of soil
properties is also important. In order to quantify these processes and predict the soil
characteristics evolution through time, dynamic process based models are required [*Hoosbeek*
*and Bryant*, 1992]. These mechanistic process models predict soil properties using both
geomorphological attributes and various physical processes such as weathering, erosion, and
bioturbation [*Minasny and McBratney*, 1999]. ARMOUR developed by *Sharmeen and*
*Willgoose* [2006] is one of the earliest process based pedogenesis models. ARMOUR
simulated surface armouring based on erosion and size selective entrainment of sediments
driven by rainfall events and overland flow, physical weathering of the soil particles which
breakdown the surface armour layer. However, very high computational resource
requirements and long run times prevented ARMOUR from performing simulations beyond
short hillslopes. Subsequently *Cohen et al.* [2009] developed mARM by implementing a
state-space matrix methodology to simplify the process based equations and calibrated its
process parameters using the results from ARMOUR. Its high computational efficiency
allowed mARM to explore the soil evolution characteristics on spatially distributed
landforms. Through their simulations *Cohen et al.* [2009] found a strong relationship between
the geomorphic quantities contributing area, slope, and soil surface grading $d_{50}$. Both
ARMOUR and mARM used two soil layers to simulate the surface armour layer and a semi-
infinite subsurface soil layer which supply sediments to the upper armour layer. For this
reason both of these models were incapable of exploring the evolution of the subsurface soil
profiles. To overcome this limitation *Cohen et al.* [2010] developed mARM3D by
incorporating multiple soil layers into mARM modelling framework. To generalise the work
of *Cohen et al.* [2010], *Welivitiya et al.* [2016] developed a new pedogenesis model called
SSSPAM, which was based on the approach of mARM3D and showed that the area-slope-$d_{50}$
relationship in *Cohen et al.* [2009] was robust against changes in process and climate
parameters and that the relationship is also true for all the subsurface soil layers, not just the
surface. Although these models predict the properties of the soil profile at an individual pixel,
they do not model the spatial interconnectivity between different parts of the soil catena
resulting from transport-limited erosion and deposition. Lateral material movement and
particle redistribution through deposition is very important in determining the soil





characteristics such as soil depth and soil texture [*Chittleborough*, 1992; *Minasny and*
*McBratney*, 2006]. In order to correctly predict the spatially distributed soil attributes and
determine the changes in soil attributes with time, coupling soil profile evolution with
landform evolution is important.
The first attempt on integrating soilscape evolution with landform evolution was done
by *Minasny and McBratney* [1999; 2001]. They used a single layer to model the influence of
soil and weathering processes on landform evolution. In addition to *Minasny and McBratney*
[1999; 2001] there are a number of conceptual frameworks found in literature for developing
coupled soil profile-landform evolution models [*Sommer et al.*, 2008; *Yoo and Mudd*, 2008].
MILESD [*Vanwalleghem et al.,* 2013] is a model which can simulate soil profile evolution
coupled with landform evolution. MILESD is built upon the conceptual framework of
landscape-scale models for soil redistribution by *Minasny and McBratney* [1999; 2001] and
pedon-scale soil formation model developed by *Salvador-Blanes et al.* [2007]. In MILESD
the soil profile is divided into four layers containing the bottommost bedrock layer and 3 soil
layers above it representing the A, B, and C soil horizons. MILESD was used to model soil
development over 60,000 years for a field site in Werrikimbe National Park, Australia
[*Vanwalleghem et al.*, 2013]. They matched trends observed in the field such as the spatial
variation of soil thickness, soil texture and organic carbon content. A limitation of MILESD
is that it only uses three layers to represent the soil profile. Recently the soil evolution
module used in MILESD has been modified to incorporate additional layers and has been
combined with the landform evolution model LAPSUS to develop a new coupled soilscape-
landform evolution model, LORICA [*Temme and Vanwalleghem*, 2015].They found similar
results for soil-landform interaction and evolution similar to MILESD simulation results.
Since only three layers were used in MILESD the representation of the particle size
distribution down the soil profile was limited. Although LORICA incorporated additional soil
layers into the MILESD modelling framework, detailed exploration of soil profile evolution
or interactions between landform evolution and soil profile evolution has not yet been done
with this model. Importantly, particle size distribution of the soil can be used as a proxy for
various soil attributes such as the soil moisture content [*Arya and Paris*, 1981; *Schaap et al.*,
2001]. The main objective of this paper is to present a new soilscape evolution model capable
of predicting the particle size distribution of the entire soil profile by integrating a previously
developed pedogenesis model in to a landscape evolution model.



151   In previous papers we have presented a pedogenesis model (on a fixed elevation

152 landform) called State Space Soil Production Assessment Model (SSSPAM) [*Welivitiya et*

153 *al.*, 2016] and explored relationships between the geomorphic parameters slope, contributing

154 area and the soil grading distribution. Similar to previous pedogenesis models such as

155 mARM3D [*Cohen et al.*, 2009; 2010], SSSPAM did not consider the interconnectivity

156 between evolving soil pedons through fluvial processes, no landform evolution was modelled

157 and no changes in the contributing area and slope occurred. In this paper we present the

158 methodology for incorporating sediment transport, deposition and elevation changes of the

159 landform in to SSSPAM modelling framework to create a coupled soilscape-landform

160 evolution model. Detailed information regarding the development and testing of SSSPAM

161 pedogenesis model is provided in previous papers by the authors ([*Cohen et al.*, 2010;

162 *Welivitiya et al.*, 2016]). The main focus of this paper is to incorporate landform evolution

163 into the SSSPAM framework. In addition to the model development we also present the

164 initial results of coupled soilscape-landform evolution exemplified on a linear hillslope.

165 **2. Model development.**

166   The introduction of a landform into the SSSPAM framework is done using a digital

167 elevation model. The structure of the landform evolution model follows that for transport-

168 limited erosion [*Willgoose et al.*, 1991] but modified so as to facilitate its coupling with the

169 soilscape pedogenesis model SSSPAM described in [*Welivitiya et al.*, 2016]. Here a regular

170 square grid digital elevation model was used and converted it into a two dimensional array

171 which can be easily processed and analysed in the Python/Cython programing language.

172 Using the "steepest-slope" criteria [*Tarboton*, 1997] the flow direction and the slope value of

173 the each pixel was determined. Then using the created flow direction matrix, the contributing

174 area of each pixel was determined using the "D8" method [*O'Callaghan and Mark*, 1984]

175 with a recursive algorithm.

176   The soil profile evolution of each pixel is determined using the interactions between

177 the soil profile and the flowing water at the surface. Figure 1 shows these layers and their

178 potential interactions. This is similar to the schematic for the standalone pedogenesis model

179 but is different in that the erosion/deposition at the surface is a result of the imbalance

180 between upslope and downslope sediment transport. The water layer acts as the medium in

181 which soil particle entrainment or deposition occurs depending on the transport capacity of

182 the water at that pixel. The water provides the lateral coupling across the landform, by the





sediment transport process. The soil profile is modelled as several layers to reflect on the fact
that the soil grading changes with soil depth depending on the weathering characteristics of
soil. Erosion of soil and/or sediment deposition occurs at the surface soil layer (surface
armour layer).
SSSPAM uses the state-space matrix approach to evolve the soil grading through the
soil profile. The state-space matrix methodology used for soilscape evolution is presented in
detail elsewhere [*Cohen et al.*, 2009; 2010; *Welivitiya et al.*, 2016] and will not be discussed
in detail here. Using this method a range of processes (e.g. erosion, weathering, deposition)
can be represented and applied so that the total change of soil layers and their properties can
be determined [*Cohen et al.*, 2009; 2010]. Once the erosion and deposition mass is
determined, the elevation changes are calculated and the digital elevation model was
modified accordingly. Once the algorithm completes modifying the digital elevation model
matrix, the calculation of flow direction and contribution area is done and the process is
repeated until a given number of iterations (evolution time) is reached.
**2.1 Characterizing erosion and deposition.**
As described in *Welivitiya et al.* [2016], the SSSPAM pedogenesis model used an
detachment-limited erosion model to calculate the amount of erosion. In order to simulate
deposition and to differentiate between erosion and deposition, a transport-limited model is
incorporated into the pedogenesis model SSSPAM. Before calculating the erosion or
deposition at a pixel (i.e. grid cell/node) we determine the transport capacity of the flow at
that particular pixel. The transport capacity determines if the pixel is being subjected to
erosion or deposition. The calculation of the transport capacity at each pixel is done
according to the empirical equation presented by *Zhang et al.* [2011] which was determined
by their flume scale sediment detachment experiments. The transport capacity at a pixel
(node) $T_c$ (kg/s) is given by,
$$T_c = \left( K_1 Q^{\delta_1} S^{\delta_2} d_{50_a}^{\delta_3} \right) \omega \qquad\qquad (1)$$
where $Q$ is the discharge per unit width (m³/s/m) at the pixel, $S$ is the slope gradient (m/m)
and $d_{50_a}$ is the median diameter of the sediment load in the flow (m), $K_1$, $\delta_1$, $\delta_2$, $\delta_3$ are
constants determined empirically and $\omega$ is the flow width (m) at the pixel. $Q$ is

    $$Q = \frac{rA_c}{\omega} \qquad\qquad (2)$$





where $r$ is runoff excess generation (m$^3$/s/m$^2$) and $A_c$ is contributing area (m$^2$) of that pixel.
Using their flume particle detachment experiments *Zhang et al.* [2011] determined that
$K_1$ =2382.32, $\delta_1$ =1.269, $\delta_2$ =1.637, and $\delta_3$ =-0.345 gave the best fit to their experimental
results. If $\underline{\psi}_{in}$ is the mass vector of the incoming sediment to the pixel, then
$L_{in} = \sum(\psi_{in_1}, \psi_{in_2} \ldots \ldots \ldots \psi_{in_n})$ (where $\psi_{in_1}, \psi_{in_2} \ldots \ldots \ldots \psi_{in_n}$ are the elements of
incoming sediment mass vector $\underline{\psi}_{in}$) is the total mass of incoming sediments to that pixel
transported by water. Using this method, SSSPAM can model the total mass of the eroded
sediment as well as the grading of the eroded material (note that elements of incoming
sediment mass vector, $\psi_{in_j}$ represents the sediment grading of the particle size class $j$).
Depending on the total incoming sediment load at the pixel, $L_{in}$, the transport capacity $T_c$ of
the flow and the potential total erosion mass $E_p$, the amount of actual erosion $E_a$ (kg/s) or
deposition $D$ (kg/s) can be determined according to Table 1. Here $\underline{\psi}_{in}$ represents the
cumulative outflow sediment mass vectors of upstream pixels $\left(\sum \underline{\psi}_{out}\right)$ which drain into the
pixel in question and is determined using the flow direction matrix mentioned earlier. The
scenario (A) and (B) (in Table 1) leads to erosion and armouring while scenario (C) leads to
deposition.
**2.2 Erosion, armouring and soil profile restructuring**
The calculation of potential erosion $E_p$ and armouring of the soil surface is done as in
*Welivitiya et al.* [2016] and *Cohen et al.* [2009]. The actual erosion $E_a$ is then determined by
adjusting the potential erosion $E_p$ according to scenarios A or B (Table 1). When calculating
the actual erosion $E_a$ we determine only the total mass of the erodible material (although it
should be remembered that total erosion is a function of the transport capacity and that is a
function of the grading $d_{50}$). The actual erosion mass vector $\underline{G}_e$ is determined using the total
soil surface mass grading vector $\underline{G}$ and erosion transition matrix **A**. The method utilized to
generate this erosion transition matrix **A** is identical to that described in detail in *Welivitiya et*
*al.* [2016] and *Cohen et al.* [2009] and will not be discussed in detail here. Briefly, the
methodology is a size selective entrainment of soil particles from the surface due to erosion
leaving the surface armour layer enriched with coarser material. It is similar to the approach
of *Parker and Klingeman* [1982] which *Willgoose and Sharmeen* [2006] showed was the best
fit to their field data for their ARMOUR surface armouring model. The eroded material is





added to the sediment load flowing into the pixel and can be given as the outflow sediment
mass vector $\underline{\psi}_{out}$.
$$\underline{\psi}_{out} = \underline{\psi}_{in} + \underline{G}_e \qquad (3)$$
The actual depth of erosion $\Delta h_E$ (m) is calculated using the equation,

$$\Delta h_E = \frac{E_a}{R_x R_y \rho_s} \qquad (4)$$


where $R_x$ and $R_y$ are the grid cell dimensions (m) in the two cardinal direction (pixel
resolution), and $\rho_s$ is the bulk density of the soil material (kg/m$^3$).

As described by the above equations, mass is removed from the surface armour layer

into the water flowing above. In SSSPAM, mass conservation of the surface armour layer is
achieved by adding a portion of soil from the 1$^{st}$ subsurface layer to the surface armour layer
equal to the mass entrained into the water flow. This material resupply propagates down the
soil profile (one soil layer supplying material to the layer above and receiving material from
the layer below) all the way to the bedrock layer which is semi-infinite in thickness. Since the
soil grading of different layers are different to each other, this flux of material through the
soil profile changes the soil grading of all the subsurface layers. Conceptually the position of
the modelled soil column moves downward since all vertical distances for the soil layers are
relative to the soil surface. In the case of deposition the model space would move upwards
(discussed in detail later). This movement of the "soil model-space" during erosion is
illustrated in Figure 2.

Note that erosion is limited by the imbalance between sediment transport capacity and

the amount of the sediment load in the flow as well as the threshold diameter of the particle
which can be entrained (Shield shear threshold, see *Cohen et al.* [2009] for details) by the
water flow. These factors limit the potential erosion rate at a pixel. During the test
simulations presented later in this paper, the depth of erosion $\Delta h_E$ was always less than the
surface armour layer thickness $D_{sur}$ (Figure 2(a)) and the rearrangement of the soil grading of
all the layers were straightforward.  However in the case of deposition, the deposition height
$\Delta h_D$ can exceed the surface armour layer thickness (and even the thickness of several soil
layers, illustrated in Figure 2(b2), (c2), if the timestep is large) and the restructuring of the





soil layer grading can be complicated. One solution to this problem is to use a smaller
timestep. But we preferred to use a conceptualization that does not impact as much on the
numerical efficiency. Details on restructuring the soil column under deposition are given in
the following section.

### 2.3 Sediment deposition

If the total mass of incoming sediment $L_{in}$ is higher than the transport capacity of the
sediment transport capacity $T_c$ at the pixel (Table 1, Scenario C) deposition of sediments
occurs at the pixel. The mass of deposited material is the difference between $L_{in}$ and $T_c$.
Although calculating the total mass of sediment which needs to deposit at a pixel ($D$) is
straightforward, determining the distribution of the deposited sediments in the form of
deposition mass vector $\underline{\Phi}$ is somewhat complicated. The deposition mass vector $\underline{\Phi}$ depends
on the size distribution of the incoming sediments which in turn depend on the erosion
characteristics of the upstream pixels. The calculation of the deposition mass vector $\underline{\Phi}$ is
done using the deposition transition matrix $\mathbf{J}$. Here $\underline{\Phi}$ is defined as,

$$\underline{\Phi} = \frac{\underline{\psi_{in}}\,\mathbf{J}}{\sum J_{z,z}\,\psi_z}D + \underline{K} \tag{5}$$


where $J_{z,z}$ are the diagonal entries of $\mathbf{J}$ (here and after the subscript $z$ denotes the $z^{\text{th}}$ grading
class), and $\psi_z$ are the elements of $\underline{\psi_{in}}$. $\underline{K}$ is an adjustment vector which modifies the values in
deposition mass vector $\underline{\Phi}$ such that $\Phi_z \le \psi_z$, where $\Phi_z$ being the elements of the vector $\underline{\Phi}$.
The adjustment vector $\underline{K}$ ensures that deposited material from each size class is not greater
than the total amount of sediment load available in the incoming sediment flow and is
iteratively determined within the deposition module of SSSAPM. The deposition of material
from the incoming sediment flow reduces the total mass of the sediment load in the flow and
changes its distribution due to this size selective deposition (particles with higher settling
velocity deposit faster). The outflow sediment mass vector $\underline{\psi_{out}}$ is then calculated by,

$$\underline{\psi_{out}} = \underline{\psi_{in}} - \underline{\Phi} \tag{6}$$

Also the deposition height $\Delta h_D$ is calculated using,




$$\Delta h_D = \frac{D}{R_x R_y \rho_s} \tag{7}$$


The following section describes the methodology for deriving the deposition transition
matrix.

### 2.3.1 Derivation of deposition transition matrix


The deposition transition matrix is derived by considering the particle trajectories at
the pixel level. Assuming all the sediments flowing into the pixel are homogeneously
distributed throughout the water column, we define the critical immersion depth $h_{ct_{(z)}}$ for all
the particle size classes as illustrated with Figure 3. The critical immersion depth is the
vertical distance travelled by the particle at the average settling velocity of the particle size
class $V_z$ where it will travel the horizontal distance of the pixel width $X$ under the flow with
the fluid flow velocity $V_f$ and settle at the far edge (i.e. exit) of the pixel.

$$h_{ct_{(z)}} = \frac{X}{V_f} V_z \tag{8}$$

Depending on the position of the sediment particle entering into the pixel with respect
to critical immersion depth, whether or not that particle will deposit in that pixel can be
determined. Particles entering to the pixel below the critical immersion depth will settle
within the current pixel, while particles entering above the critical immersion depth will stay
in suspension and exit the current pixel. The critical immersion depth is greater for larger (or
more dense) particles and less for smaller (or less dense) particles. For sediment particles in
larger size classes, the critical immersion depth can be larger than the flow depth $H_f$ (m)
(thickness of the water column). That means all the particles in that particle size class will
settle in the pixel. Using the critical immersion depth and the flow depth we can define the
diagonal elements $J_{z,z}$ of the deposition transition matrix $\mathbf{J}$ in following manner.
$$J_{z,z} = \begin{cases} \frac{h_{ct_{(z)}}}{H_f} & for \ H_f \geq h_{ct_{(z)}} \\ 1 & for \ H_f < h_{ct_{(z)}} \end{cases} \tag{9}$$
Note the deposition transition matrix $\mathbf{J}$ is a diagonal matrix which contains only
diagonal elements (all off diagonal elements being 0). The evaluation of elements in the

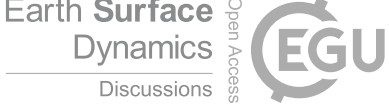

potential deposition matrix **J** requires the calculation of the critical immersion depth $h_{ct_{(z)}}$ and
the flow depth $H_f$.

The following discussion briefly describes the methodology used to calculate the

above variables. The average settling velocity of all the particle sizes classes can be
calculated for typical sediment sizes using Stoke's Law [*Lerman*, 1979].

$$V_z = \frac{(\rho_s - \rho_f)g}{18\mu}d_z^2 \qquad (10)$$


where $\rho_s$ and $\rho_f$ are bulk density of the soil particles and the density of water (kg/m$^3$) (fluid),
$g$ is gravitational acceleration (m/s$^2$), $d_z$ is the median particle diameter of the size class $z$ (m)
and $\mu$ is the dynamic viscosity of water (kg/s/m$^2$). The average flow velocity and the flow
depth can be calculated using the Manning formula [*Meyer-Peter and Müller*, 1948;
*Rickenmann*, 1994]. Although the Manning formula is normally used to calculate the average
flow velocity in channels, we assume that the same formula can be used to calculate the flow
velocity at the pixel level assuming water flowing over a pixel as a small channel segment.
Manning formula states,
$V_f = \frac{1}{n}R^{2/3}\,S^{1/2}$ \qquad (11)
where $n$ is the Manning's roughness coefficient, $R$ is the hydraulic radius (m) and $S$ is the
slope (m/m). The Manning's roughness coefficient $n$ can be approximated using the median
diameter $d_{50}$ (mm) of the surface armour layer [*Coon*, 1998] using following equation.
$n = 0.034(d_{50})^{1/6}$ \qquad (12)

The hydraulic radius is the ratio between the cross-sectional area of the flow and the

wetted perimeter. When we consider the flowing water column at a pixel, the cross-sectional
area of the flow is the multiplication of flow width (pixel width) $\omega$ and the flow depth $H_f$
with the wetted parameter being the flow width $\omega$. The hydraulic radius at the pixel is then
the flow depth $H_f$. Substituting flow depth for hydraulic radius equation (11) becomes,

$$V_f = \frac{1}{n}H_f^{2/3}\,S^{1/2} \qquad (13)$$




The flow velocity at the pixel can be also expressed in terms of upslope contributing
area $A_c$ , runoff excess generation $r$, flow width $\omega$ and flow depth $H_f$.

$$V_f = \frac{A_c r}{H_f \omega} \tag{14}$$

Solving the equations (13) and (14) the flow depth $H_f$ and the flow velocity $V_f$ can be
calculated in terms of $A_c, r, \omega, S$ and $n$ using

$$H_f = \left(\frac{A_c r n}{\omega S^{1/2}}\right)^{3/5} \tag{15}$$

$$V_f = \left(\frac{A_c q}{l_c}\right)^{2/5} \left(\frac{S^{3/2}}{n^3}\right)^{1/5} \tag{16}$$

**2.3.2 Restructuring of the soil layers after deposition**
Deposition of sediment on the soil surface moves the soil surface upwards (soil
model-space moves upwards). As mentioned earlier the deposition height $\Delta h_D$ can exceed the
surface armour layer thickness and/or a number of subsurface soil layer thicknesses. Figure
2(b2) illustrates a typical scenario where the deposition height has exceeded the thickness of
the surface armour layer $D_{sur}$.
Figure 2(b2) and (c2) shows the movement of the model-space for three soil layers. In
the restructured soil column (Figure 2(c2)) the new 3$^{rd}$ layer consists of a portion of the
original layer one (surface armour layer) and the 1$^{st}$ original subsurface layer. Because of the
upward movement of the model-space, a portion of the 2$^{nd}$ original soil layer and the entire
3$^{rd}$ soil layer has been incorporated into the new bedrock layer. However, the grading of the
new bedrock layer remains unchanged although the material from the original soil layers two
and three is added to the bedrock layer. At the first glance it may seem that this process
would drastically alter the soilscape evolution dynamics by introducing a sharp contrast in
soil grading at the soil-bedrock interface. In SSSPAM a large number of soil layers (50 to
100) are used to ensure smooth soil grading transition from soil to bedrock.
Figure 4 shows three different cases that can occur during the deposition process. Let
the soil grading mass vector of the original soil surface be $\underline{G}_{sur}$ and $\underline{G}_{sub(1)}$, $\underline{G}_{sub(2)}$, ..........,
$\underline{G}_{sub(n)}$ be the soil grading mass vectors of the original subsurface layers. In the same manner





367 let $\underline{G}''_{sur}$ be the soil grading mass vector of the new surface armour layer and $\underline{G}''_{sub(1)}$, $\underline{G}''_{sub(2)}$,

368 ..., $\underline{G}''_{sub(n)}$ be the soil grading mass vectors of the new subsurface layers, and $D_{sur}$ and $D_{sub}$

369 are the thickness of surface armour layer and the thickness of each subsurface layer

370 respectively. Depending on the position of the original surface armour layer in the new soil

371 column, different approaches need to be taken in order to calculate the new soil gradings as

372 described in following cases.

373 **Case 1:**

374  In Case 1 (Figure 4(b)) the deposition height $\Delta h_D$ is less than the surface armour

375 thickness $D_{sur}$. Considering the uniform soil column cross-sectional area, the new soil layer

376 mass grading vectors of different soil layers (for Case 1) are calculated as,

$$\underline{G}''_{sur} = \underline{\Phi} + \left(\frac{D_{sur} - \Delta h_D}{D_{sur}}\right)\underline{G}_{sur} \tag{17}$$

$$\underline{G}''_{sub(1)} = \left(\frac{\Delta h_D}{D_{sur}}\right)\underline{G}_{sur} + \left(\frac{D_{sub} - \Delta h_D}{D_{sub}}\right)\underline{G}_{sub(1)} \tag{18}$$

$$\underline{G}''_{sub(i)} = \left(\frac{\Delta h_D}{D_{sur}}\right)\underline{G}_{sub(i-1)} + \left(\frac{D_{sub} - \Delta h_D}{D_{sub}}\right)\underline{G}_{sub(i)} \tag{19}$$

379 where $i$ is the number of new subsurface soil layers such that $i \in \{2, 3, ....., n\}$ and $n$ is the

380 number of subsurface layers.

381 **Case 2:**

382  In Case 2 (Figure 4(c)) the deposition height $\Delta h_D$ is greater than the surface armour

383 layer thickness $D_{sur}$ and the original surface armour layer is situated inside a single new

384 subsurface layer. Also the new soil subsurface layer which contains the original surface

385 armour layer can reside in any depth within new soil profile depending on the deposition

386 height (e.g. it can be 1st, 2nd, 5th or any subsurface layer). For simplicity of explanation Figure

387 4(c) shows this layer being in the 1st new subsurface layer. In the model the original surface

388 armour layer is contained in the $k$th new subsurface layer. In this instance the new surface

389 armour layer and all the new subsurface layers above $k$th layer will have the same particle size

390 distribution as the deposition mass vector $\underline{\Phi}$. They are (using the same notation as before),


$$\underline{G}''_{sur} = \left(\frac{D_{sur}}{\Delta h_D}\right)\underline{\Phi} \tag{20}$$






$$\underline{G}^{''}_{sub(i)} = \left(\frac{D_{sub}}{\Delta h_D}\right)\underline{\Phi} \tag{21}$$

where $i \in \{1, 2, \ldots, k\text{-}1\}$.

Case 2 satisfies the condition $kD_{sub} \geq \Delta h_D > D_{sur}$, when the original surface armour

layer belongs to a single subsurface layer. The $k^{th}$ new subsurface layer contains the
contribution from three different sources, (1) deposited material, (2) material from the
original surface armour layer, and (3) material from the original $1^{st}$ subsurface layer. Using
the soil grading mass vectors of these sources the soil grading mass vector of the $k^{th}$
subsurface layer is,

$$\underline{G}^{''}_{sub(k)} = \left(\frac{\Delta h_D - D_{sur} - (k-1)D_{sub}}{\Delta h_D}\right)\underline{\Phi} + \underline{G}_{sur} + \left(\frac{kD_{sub} - \Delta h_D}{D_{sub}}\right)\underline{G}_{sub(1)} \tag{22}$$


The soil grading mass vectors of all the other new subsurface layers is,

$$\underline{G}^{''}_{sub(i)} = \left(\frac{\Delta h_D - (k-1)D_{sub}}{D_{sub}}\right)\underline{G}_{sub(i-k)} + \left(\frac{kD_{sub} - \Delta h_D}{D_{sub}}\right)\underline{G}_{sub(i-k+1)} \tag{23}$$


where $i \in \{k\text{+}1, k\text{+}2, \ldots, n\}$.
**Case 3:**

Calculation of the soil grading mass vectors for the Case 3 (Figure 4(d)) is similar to

Case 2. In this case the deposition height $\Delta h_D$ is greater than the surface armour layer
thickness $D_{sur}$ and the original surface armour layer belongs to two new subsurface layers.
As was with Case 2, the new soil subsurface layers, which contain portions of the original
surface armour layer, can reside at any depth within the new soil profile. Figure 4(d) shows
the situation where the surface layer now resides in both $1^{st}$ and $2^{nd}$ new subsurface layers.
The model assumes that the original surface armour layer is contained in both $k^{th}$ and $k\text{+}1^{th}$
new subsurface layers. Similar to Case 2 the new surface armour layer and all the new
subsurface layers above $k^{th}$ layer will have the same particle size distribution as the
deposition mass vector $\underline{\Phi}$ and they are calculated using the same equations (20) and (21).





Case 3 (Figure 4(d)) satisfies the condition $(D_{sur} + kD_{sub}) \geq \Delta h_D > kD_{sub}$. The new $k^{th}$ subsurface layer contains the contribution from the deposited material and the material from the original surface armour layer while $k+1$ layer containing contributions from the original surface armour layer and the first original subsurface layer. The soil grading mass vectors for new $k^{th}$ layer and $k+1^{th}$ layer are,

$$\underline{G}''_{sub(k)} = \left(\frac{\Delta h_D - D_{sub} - (k-1)D_{sub}}{\Delta h_D}\right)\underline{\Phi} + \left(\frac{D_{sub} + kD_{sub} - \Delta h_D}{D_{sub}}\right)\underline{G}_{sur} \qquad (24)$$

$$\underline{G}''_{sub(k+1)} = \left(\frac{\Delta h_D - kD_{sub}}{D_{sub}}\right)\underline{G}_{sur} + \left(\frac{(k+1)D_{sub} - \Delta h_D}{D_{sub}}\right)\underline{G}_{sub(1)} \qquad (25)$$

The soil grading mass vectors of all the other new subsurface layers is calculated by,

$$\underline{G}''_{sub(i)} = \left(\frac{\Delta h_D - kD_{sub}}{D_{sub}}\right)\underline{G}_{sub(i-k-1)} + \left(\frac{(k+1)D_{sub} - \Delta h_D}{D_{sub}}\right)\underline{G}_{sub(i-k)} \qquad (26)$$

where $i \in \{k+2, k+3, \ldots, n\}$

## 2.4 Soil profile weathering

The methodology used for simulating the weathering within the soil profile is detailed by *Welivitiya et al.* [2016]. It used a physical fragmentation mechanism where a parent particle disintegrates into $n$ number of daughter particles where a single daughter particle retaining $\alpha$ fraction of the parent particle by volume and the remaining daughter particles retaining $1 - \alpha$ fraction of the parent particle volume. By changing $n$ and $\alpha$ we can simulate a wide range of particle disintegration geometries which can be attributed to different weathering mechanisms. In this paper we used $n = 2$ and $\alpha = 0.5$ to simulate symmetric fragmentation mechanism where a single parent particle breaks down in to 2 equal daughter particles. But the model can simulate any values of $n$ and $\alpha$. We decided to use the symmetric fragmentation mechanism based on the results of *Wells et al.* [2008]. Using the above mentioned parameters, parent - daughter particle diameters and soil grading distribution values, the weathering transition matrix is constructed according to the methodology described by *Cohen et al.* [2009] and will not be discussed further.

The weathering rate of each soil layer is simulated using a depth dependent weathering function. It defines the weathering rate as a function of the soil depth relative to

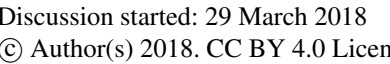



the soil surface depending on the mode of weathering of that particular material. SSSPAM is
capable of using different depth depending weathering functions to simulate the soil profile
weathering rate. For the initial simulations presented in this paper we used the exponential
[*Humphreys and Wilkinson*, 2007] and humped exponential [*Ahnert*, 1977; *Minasny and*
*McBratney*, 2006] depth dependent weathering functions. Detailed explanation and the
rationale of these weathering functions is presented in *Welivitiya et al.* [2016] and extended
by Willgoose [2018].

**3 SSSPAM simulation setup**

The objective of the simulations below was to explore the capabilities and
implications of the SSSPAM coupled soilscape-landform evolution model. Although the
model is capable of simulating soilscape and landform evolution for a three-dimensional
catchment scale landform, a synthetic two-dimensional linear hillslope (length and depth)
landform was used here. Because it is two-dimensional, the landform always discharges in a
single direction. In this way the complexities of multidirectional discharge were avoided so
we can focus on the soilscape-landform coupling.
Figure 5 shows the synthetic landform used. The simulated landform starts from an
almost flat 1 km long plateau (almost flat area at the top of the hillslope) with a very small
gradient of 0.001%. A hillslope with a gradient of 2.1% starts at the edge of the plateau and
continues 1.5 km horizontally while dropping 31.5 m vertically and terminates at a valley.
The valley (another almost flat area at the bottom of the hillslope) itself has the same gradient
as the upslope plateau (0.001%) and continues for another 1 km. The valley (the bottom
section of the landform) is designed to facilitate sediment deposition so the effect of sediment
deposition on soilscape development can be analysed. The simulated hillslope has a constant
width of 10 m (one pixel wide) and is divided into 10 m longitudinal segments with the total
number of pixels being 350. At each pixel the soil profile is defined by a maximum of 102
soil layers. The soil surface armour layer is the topmost soil layer and it has a thickness of 50
mm. The 100 layers below the surface layer are subsurface soil layers with a thickness of 100
mm each. The bottommost layer ($102^{nd}$ layer) is a permanent non-weathering layer and it is
the limit of the hillslope modelling depth. In this way SSSPAM is capable of modelling a soil
profile with a maximum thickness of 10.05 m. By changing the number of soil layers used in
the simulation SSSPAM is able to simulate a soil profile with any thickness. However as the



number of model layers increase, the time required for the each simulation also increases.
During our initial testing, we found that the soil depth rarely increased beyond 10 m and
decided to set 10.05 m as the maximum soil depth for this initial scenario.

Two soil grading data sets (Table 2) were used for the initial surface soil grading and
the bedrock. The first soil grading was from Ranger Uranium Mine (Northern Territory,
Australia) spoil site. This soil grading was first used by *Willgoose and Riley* [1998] for their
landform simulations. It was also subsequently used by *Sharmeen and Willgoose* [2007] for
their work with ARMOUR simulations and *Cohen et al.* [2009] for mARM simulation work.
The soil grading consisted of stony metamorphic rocks produced by mechanical weathering
with a body fracture mechanism [Wells et al., 2008]. It had a median diameter of 3.5 mm and
a maximum diameter of 19 mm (Table 2 - Ranger1a). The second grading was created to
represent the bedrock using the size classes of the previous soil grading. It contained 100% of
its mass in the largest particle size class that is 19 mm (Table 2 - Ranger1b). These soil
gradings are the same soil gradings used in the SSSPAM parametric study of *Welivitiya et al.*
[2016]. At the start of the simulation the surface armour layer was set to the soil grading
(Table 2 - Ranger1a) and all the subsurface layers were set to bedrock grading (Table 2 -
Ranger1b). The discharge (runoff excess generation) rate of water is derived from averaging
the 30 year rainfall data collected by *Willgoose and Riley* [1998]. Using the simulation setup
described above simulations was carried out using the yearly averaged discharge rate.

**4 Simulation results with exponential weathering function**

Figure 6 shows six outputs at different times during hillslope and soil profile
evolution.

The upper section in each of the panels in Figure 6 is the cross-section median
diameter ($d_{50}$) of the soil profile and the landform, with the line denoting the original
landform surface. The middle panel is the median diameter $d_{50}$ of the soil surface armour
layer. The bottom panel is the soil profile relative to the surface highlighting the soil profile
$d_{50}$. The soil depth is the depth below the surface at which $d_{50}$ reaches the maximum possible
particle size (i.e. the bedrock grading). Figure 6(a) shows the initial condition for the
soilscape: a deep bedrock overlain by a very thin fine-grained soil armour. The evolution of
the coupled soilscape and landform at different simulation times are presented in subsequent
Figures 6(b) - 6(f).



If we initially consider the landform evolution alone, the erosion-dominated regions
and the deposition-dominated regions can be clearly identified. Initially erosion is highest on
top of the hillslope where the plateau transitions to the hillslope (plateau-hillslope boundary)
and erosion gradually reduces down the hillslope. Also, there is a sharp increase of surface
$d_{50}$ at the plateau-hillslope boundary and then a gradual decrease down the hillslope. The
transport capacity of the flow and deposition has the highest erosion (i.e. the rate of change in
the sediment transport capacity) occurring at the top of the hillslope. The summit plateau has
a very low slope gradient and although the contributing area increases across the plateau, the
potential erosion and the transport capacity of the flow remains negligible resulting in
minimum erosion. At the plateau-hillslope boundary, the slope gradient suddenly increases.
This increase in slope gradient and high contributing area increases the potential erosion of
the flow and causes a rapid increase in transport capacity downslope. This erosion gradually
reduces further down the hillslope despite increasing contributing area. Although the
transport capacity increases towards the bottom of the hillslope, water flowing over the
downslope nodes is laden with sediments already eroded from upslope nodes. This reduces
the amount of erosion at the downslope nodes.
Turning to the evolution of the soil profile, the upslope plateau retains the initial
surface soil layer without any armouring due to the very low erosion and it develops a
relatively thick soil profile as a result of bedrock weathering. The high erosion rate at the
plateau-hillslope boundary removes all the fine particles from the initial soil layer as well as
fine particles produced by weathering process, creating a very course surface armour layer.
This high erosion rate also leads to a relatively shallow soil profile. The erosion rate reduces
down the slope due to saturation of the flow with sediments from upstream. Low erosion
leads to a weak armouring and the fine particles produced from surface weathering remain on
the surface. These processes lead to the fining of the surface soil layer and thickening of the
soil profile down the hillslope.
With time the location of the high erosion region shifts upstream onto the plateau
cutting into it. The $d_{50}$ of the armour layer downslope also decreases. Both of these changes
occur due to lowering of the slope gradient of the hillslope over time.
Deposition of material occurs either side of the hillslope-valley boundary. The valley
at the foot of the hillslope has a very low initial slope gradient. At the hillslope-valley
boundary (toe slope) the slope gradient reduces suddenly. This sudden slope gradient



reduction reduces the transport capacity of the water flow and initiates deposition. Initially
deposition occurs only at the hillslope-valley boundary node and increases its elevation. This
deposition and slope reduction propagates upslope until equilibrium is reached with erosion.
Deposition propagates across the valley and produces the deposits in the Figure 6.
There is a change in the surface $d_{50}$ between the erosion and deposition regions at
around 2000 m. The surface $d_{50}$ of the erosion region reduces down the slope, reaches a
minimum at 2000 m and then increases as it transits into the deposition region. This can be
clearly seen in Figures 6(c) and 6(d). As noted previously the "actual erosion rate" reduces
down the slope due to saturation of the flow with sediments. At the end of the erosion region
no more erosion can take place because the flow is completely saturated with sediment.
Because of the lack of erosion, fine particles are not removed from the surface and
weathering produces more and more fine particles reducing the surface $d_{50}$ and increasing the
soil depth.
Near the erosion-deposition boundary only a small amount of sediment is deposited.
Since the larger particles have the highest probability of deposition, a small amount of coarse
material deposits there. Downslope into the deposition region the slope further decreases, the
difference between the transport capacity and the sediment load increases and the rate of
deposition steadily increases. Since larger particles have a higher probability of depositing
first, coarse material preferentially deposits. Mixing of these coarse particles with pre-
existing weathered fine particles produces the observed coarsening of the surface $d_{50}$. Once
the surface $d_{50}$ of the deposition region reaches a peak it starts to decrease again (from 2500
m to 3000 m). Beyond 3000 m the deposited material is smaller because the larger particles
have already been deposited upstream. The deposition of each consecutive downstream node
consists with finer particles leading to the observed decrease of surface and profile $d_{50}$. As
expected the soil thickness is higher in the deposition regions than the other regions.
With time the deposition region moves upslope. The gradient of $d_{50}$ observed in
earlier times of the deposition region (until 30,000 years) decreases and the soil changes into
a very fine-grained homogeneous material resulting from surface weathering. Due to the high
weathering rate at the surface and the upper soil layers, the deposited sediment decomposes
into a very fine material. With time, the $d_{50}$ of the sediments in the water flow also decreases
due to low erosion potential and weathering of the surface armour layer of upslope nodes. For





these reasons the $d_{50}$ of the deposition region decreases and becomes homogeneous leading to
burial of the coarse material that was deposited earlier .

**4.1 Evolution characteristics of different sites**

In order to better understand the dynamics of soilscape evolution we also plotted the
elevation, slope, rate of erosion (and/or deposition), surface $d_{50}$, soil depth and profile $d_{50}$ for
four sites (Figure 6(a)). The first two sites (sites 1 and 2) are either side of the plateau-
hillslope boundary in the erosion region The other two sites (sites 3 and 4) are either side of
the hillslope-valley boundary in the deposition region.

*Site 1 and 2:*

For site 1 (Figure 7- solid line plots) the erosion and surface $d_{50}$ are strongly
correlated over time. The soil depth and profile $d_{50}$ plots are also highly correlated. The
abrupt change in profile $d_{50}$ occurs at the same time as abrupt changes in soil depth. Site 1
initially has small erosion because the slope is very low so weathering dominates. This small
erosion means the elevation and slope are initially constant. Due to the dominance of
weathering, both surface and profile grading becomes enriched with fine particles and the $d_{50}$
decreases. Weathering of the profile layers creates a relatively deep soil profile. With time the
erosion front, initially at the plateau-hillslope transition, cuts back into the plateau. The
increased erosion rate removes the fine material created by weathering leading to a coarse-
grained armour.
When the erosion front crosses site 1, the gradient increases as does the erosion rate
(at around 20,000 years). During this phase of increasing erosion the surface $d_{50}$ also
increased. However, the surface $d_{50}$ stabilizes around 14 mm before the erosion rate reaches
its maximum value. This is because once total armouring occurs, the erosion is reduced to a
very low value. Although the erosion is low, the slope of the site 1 continues to increase until
it reaches a maximum and the Shield's shear stress threshold diameter also increases. This
allows erosion to keep increasing while the surface $d_{50}$ remain essentially constant. When the
erosion rate overtakes the rate of production of weathering, the soil depth decreases.
Increasing erosion reduces the soil thickness while coarsening the surface of upper soil
layers. This results in the increase of the profile $d_{50}$ at later times. At 20,000 years, the
reduction of slope reduces the rate of erosion so that, weathering again dominates the site.
Weathering produces more fine particles reducing the surface $d_{50}$ from about 48,000 years.



The dominance of weathering over erosion also increases the soil depth while decreasing the
profile $d_{50}$.

Both soil depth and profile $d_{50}$ plots resemble a stair-stepped graph. The reason for

this appearance is that SSSPAM calculates soil depths as the number soil profile layers. The
model doesn't interpolate the depth of soil within a single layer. Since the profile $d_{50}$ is a
function the soil thickness, this plot also displays this pattern.

For site 2 (Figure 7-dashed line plots) the evolution is simpler than site 1. The initial

transport capacity and discharge energy at site 2 is very high while the sediment inflow from
upstream is low because of low erosion from the plateau. The resulting higher erosion rate
produces a very coarse surface layer and exposes the bedrock in the subsurface. This effect
causes both the surface $d_{50}$ and profile $d_{50}$ to rapidly increase to the maximum possible
diameter (bedrock grading).

Although the surface $d_{50}$ has reached the maximum possible diameter the erosion

continues to increase as the Shield's threshold diameter for entrainment of the water flow has
increased beyond the maximum particle size (19 mm) and the bedrock grading itself is being
eroded. However, at around 2,700 years the Shield's threshold diameter decreases below 19
mm and the fully armoured surface causes the erosion rate to decrease rapidly and becomes
unstable in time with rapid fluctuations. Once an armour layer develops on the surface, the
profile layers are protected from erosion and weathering becomes more dominant, so the
profile $d_{50}$ decreases while soil depth increase.
***Site 3 and 4:***

For site 3 (Figures 8-solid line plots) the elevation increases due to deposition. The

initial increase of surface $d_{50}$ occurs due to size selective deposition. As noted in the model
description, larger particles deposit at a higher rate. This deposition of larger particles on the
surface causes the surface $d_{50}$ to initially increase.

The subsequent decrease of the surface $d_{50}$ occurs due to a combination of two

processes. Firstly, with time the upstream boundary of the deposition region moves upslope
and since the largest particles tend to deposit at the beginning of the deposition region, the
sediment flow at site 3 gets enriched with more and more fine particles. Due to the deposition
of these relatively finer particles the surface $d_{50}$ tends to decrease. Secondly, weathering of
the surface and the subsurface layers reduces the surface $d_{50}$. Compared to sites 1 and 2 the





soil depth increase of site 3 is much higher. In sites 1 and 2 the soil profile growth only
occurred due to the excess of weathering over erosion. At site 3 the soil layer grows due to
material deposition as well as weathering of the bedrock. The profile $d_{50}$ increases in the
initial stage.
For site 4 (Figures 8-dashed line plots) while the initial evolution is different, in the
latter stages (beyond year 15,000) the evolution characteristics of the soil properties are
similar to that of site 3. Since the valley initially has a low slope, the initial erosion is
negligible and the elevation, slope and erosion remain close to 0. With the growth of the
deposition region, a "deposition front" moves across the valley. Before the deposition front
reaches site 4, the elevation, slope and erosion/deposition remain unchanged.  Because the
initial erosion rate at the site 4 is low, there is no armouring so that weathering dominates and
the surface $d_{50}$ decreases.  When the deposition front reaches site 4, the elevation increases
due to sediment deposition as so does the slope. Due to the size selective deposition of coarse
sediment the surface $d_{50}$ increases. Afterwards the evolution of the soil properties is similar to
site 3 as the same processes are acting at sites 3 and 4.

## 5 Simulation results with humped exponential weathering function

To test the sensitivity of the conclusions in the previous section to changes in the
depth dependent weathering functions, in this section we explore the effect of weathering
using the humped exponential weathering function. The key difference is that the humped
function has a low weathering rate at the surface with the peak weathering rate occurring
mid-profile.
Superficially, both the humped and exponential weathering functions produce  similar
trends, however there are some differences in the particle size distribution, soil depth and the
evolution of the landform (Figure 9). At identical times the surface $d_{50}$ is coarser and the soil
depth is less for the humped simulations. There is also a subtle difference in the initial
landform evolution. For the exponential weathering function the highest erosion rate occurs
near the plateau-hillslope boundary (year 2000 near 1,000 m, Figure 6). For the humped
function this maximum soil surface deviation occurs further down the hillslope (year 2000
near 1500 m, Figure 9). For subsequent times, this difference in the location of the maximum
erosion leads to subtly different landforms.



These differences in landform evolution are explained by the near surface weathering
rates. For the exponential weathering function the weathering rate is highest at the surface
and declines exponentially with depth. For the humped exponential weathering function the
highest weathering rate is at a finite depth below the surface and exponentially decrease
below and above this depth. Because of the lower surface weathering rate for humped, the
surface $d_{50}$ remains coarser during the entire simulation. The relative coarseness of the
surface means that the water flow needs to be more energetic to entrain material from the
surface due to the Shields's stress entrainment threshold. For the exponential weathering
function simulations, shear stress of the water flow is high enough to entrain most of the
surface soil particles near the plateau-hillslope boundary owing to the finer armour layer as a
result of surface weathering. However for the humped exponential weathering simulations the
surface armour is coarser because of the lower surface weathering rate and the shear stress of
the water flow is not high enough to detach material from the armour layer. Because of this,
the highest erosion occurs downslope where the contributing area is higher and hence the
shear stress of the water flow is higher.

**6 Conclusions**
This study presented the methodology for incorporating landform evolution into the
SSSPAM pedogenesis model. This was achieved by incorporating elevation changes
produced by erosion and deposition. Previous published work with SSSPAM assumed that
the landform, slope gradients and contributing areas remained constant during the simulation.
This did not preclude the landform evolving, only that the soil reached equilibrium faster (i.e.
had a shorter response time) than the landform evolved (i.e. a "fast" soil, Willgoose, 2018). In
the new version of SSSPAM discussed here, the elevations, contributing area, slope gradient
and slope directions at each node dynamically evolve. This new model explicitly models co-
evolution of the soil and the landform, where the response time for soil and landform are
similar.
By defining "the critical immersion depth", a novel and simple methodology for size
selective deposition was introduced to formulate the deposition transition matrix. This
deposition transition matrix characterises the size selectivity of sediment deposition
depending on the settling velocity of the sediment particle, with faster settling velocity
particles settling first.



The results demonstrated SSSPAM's ability to simulate erosion, deposition and weathering processes as well as soil formation and its evolution coupled with an evolving landform. The simulation results qualitatively agree with general trends in soil catena observed in the field. The model predicts the development of thin and coarse-grained soil profile on the upper eroding hillslope and thick and fine-grained soil profile at the bottom valley. Considering the dominant process acting upon the soilscape, the hillslope can be divided into weathering-dominated, erosion-dominated and deposition-dominated sections. The plateau (summit) was mainly weathering-dominated due to its very low slope gradient and low erosion rate. The upper part of the hillslope was erosion-dominated owing to its high slope gradient and high contributing area. The lower part of the hillslope and the valley was deposition-dominated. The position and the size of these sections changes with time due to the evolution of the landform and the soil profile. During the simulation, the weathering-dominated region shrinks due to the erosional region dominating it. The erosion-dominated region expands upslope into the previously weathering-dominated region and the downstream boundary retreats upslope away from the deposition-dominated region, but shows a net expansion in area. The deposition-dominated region expands upslope into the previously erosion-dominated region with a net expansion.

The simulation results also show how the interaction of different processes can have unexpected outcomes in terms of soilscape evolution. The best example is the fining of the surface grading despite an increasing transport capacity and potential erosion rate. This occurs due to saturation of the flow with sediment eroded from upstream nodes. Further, the comparison of results produced by the exponential and humped exponential weathering functions showed how the distribution of weathering rate down the soil profile changes the overall properties of the soilscape. For instance, the humped exponential simulation produced a thinner soil profile and coarser soil surface armour compared with simulation results of exponential weathering function because of the reduced weathering rate at the soil surface. This led to a longer-lived surface armour for the humped function.

The synthetic landform simulations demonstrated SSSPAM's ability to qualitatively simulate erosion, deposition and weathering processes and to generate familiar soilscapes observed in the field. Comparison of results obtained from two different depth different functions demonstrate how the soilscape dynamic evolution is influenced by the weathering mechanisms. This in turn links to the geology of the soil parent material and their preferred weathering mechanism which leads to the heterogeneity of soilscape properties in a region. A





future paper will discuss how this work can be extended to include the impact of chemical
weathering into soilscape evolution.

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



**Figure 1**


**Figure 1** Schematic diagram of the SSSPAM model.








**Figure 2**


**Figure 2** Erosion, Deposition and the restructuring of the soil profile (a) original soil profile,
(b1, c1) for erosion, (b2, c2) for deposition.
















**Figure 3**

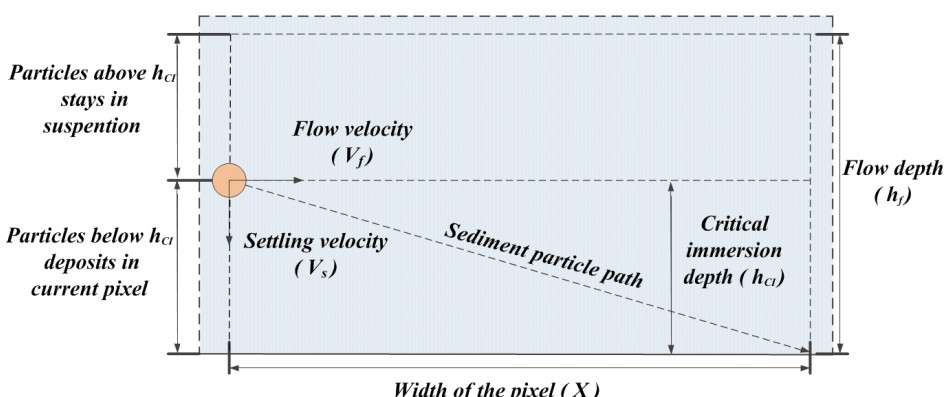


**Figure 3** Determination of critical immersion depth of a sediment particle














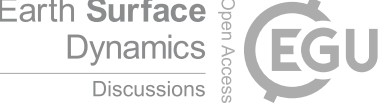


**Figure 4**

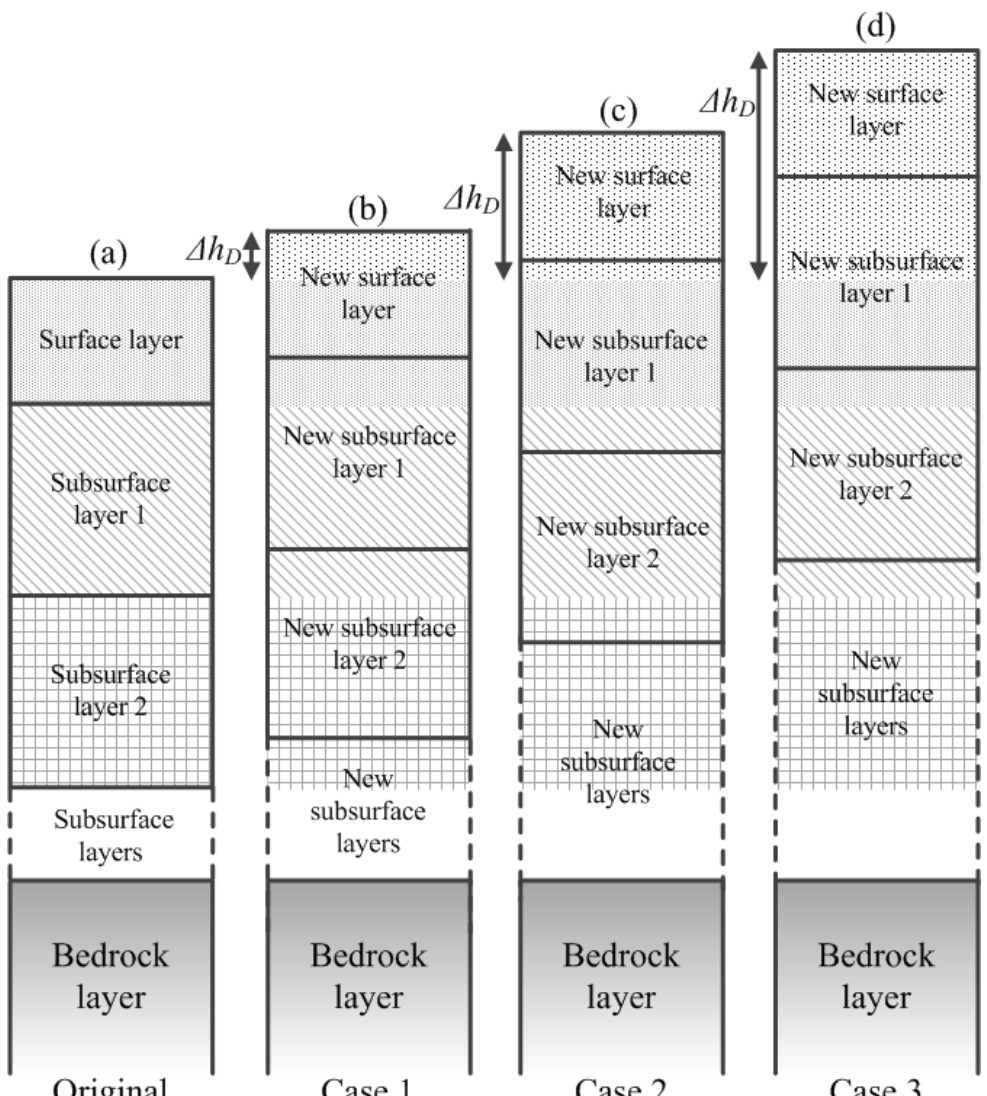


**Figure 4** Different deposition scenarios











**Figure 5**

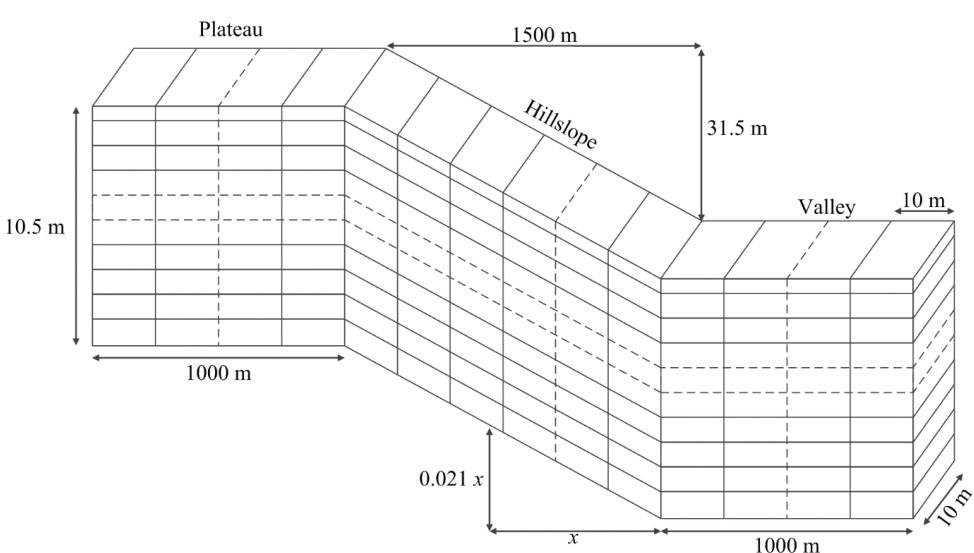


**Figure 5** The simulated landform and the definition of nodes.


















**Figure 6**


**Figure 6** Evolution of the soilscape with the exponential depth dependent weathering
function.







**Figure 7**

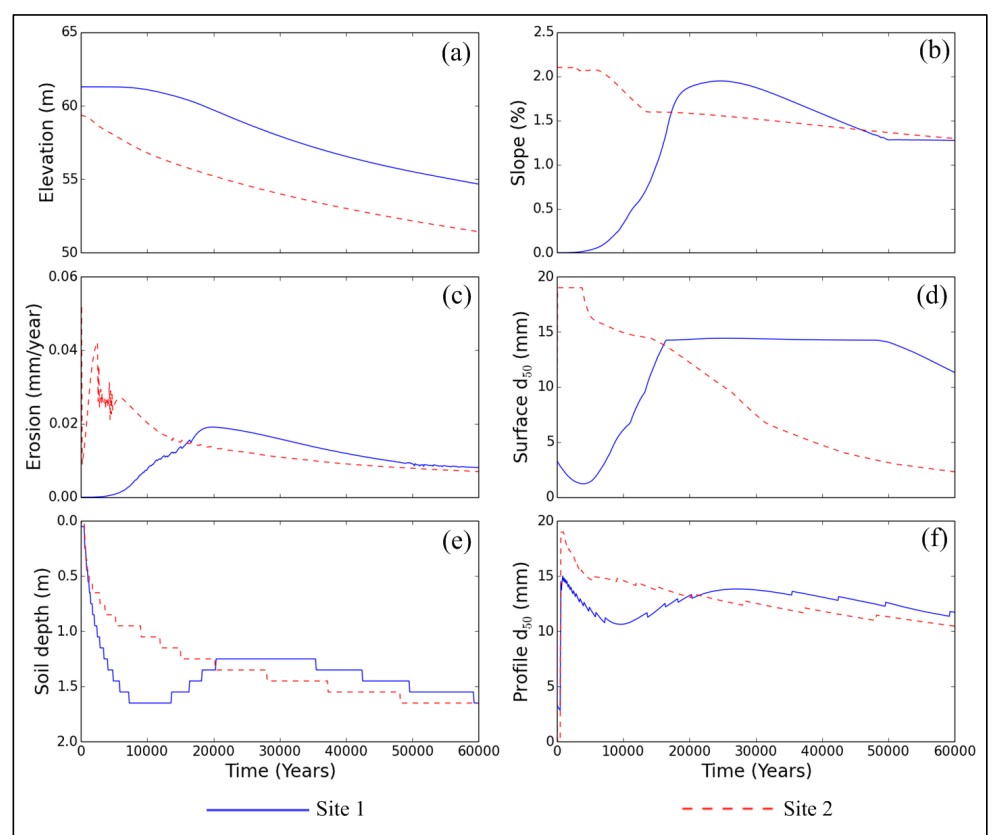


**Figure 7** Evolution characteristics of Sites 1 and 2, (a) elevation, (b) hillslope gradient, (c) erosion rate, (d) surface $d_{50}$, (e) soil depth, and (f) profile $d_{50}$.















**Figure 8**

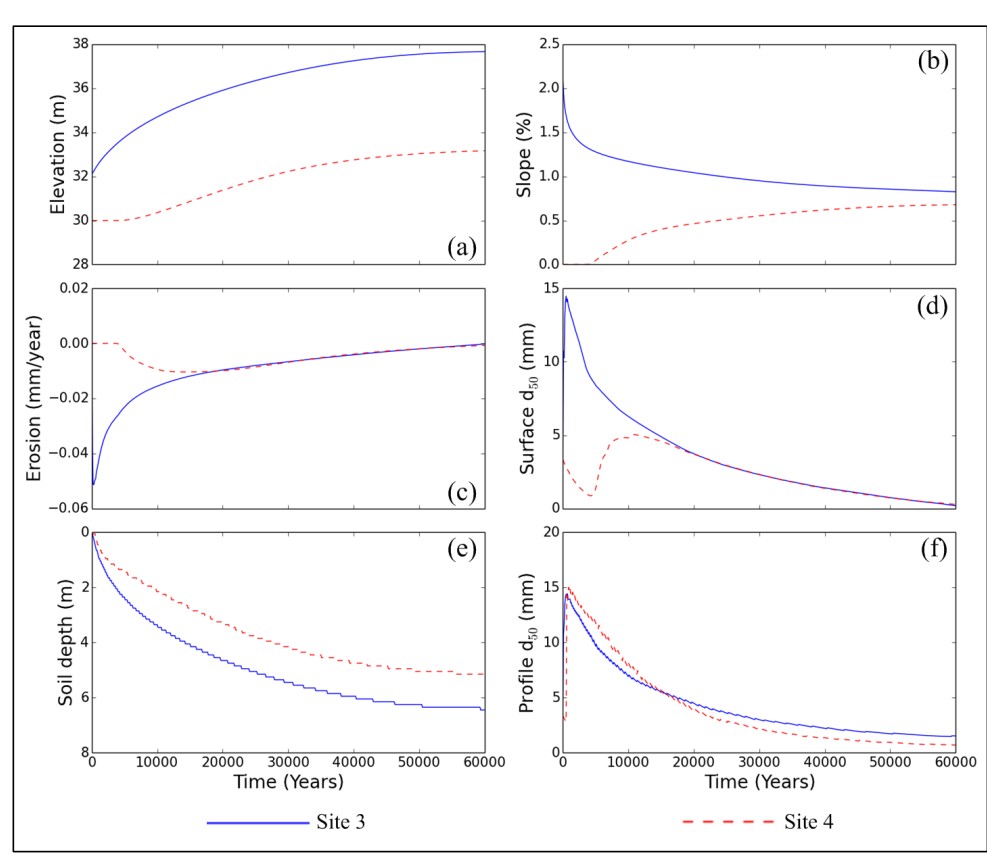


**Figure 8** Evolution (near the hillslope-valley boundary) of Sites 3 and 4, (a) elevation, (b)
hillslope gradient, (c) erosion rate, (d) surface $d_{50}$, (e) soil depth, and (f) profile $d_{50}$.














**Figure 9**



**Figure 9** Evolution of the soilscape with the humped exponential depth dependent weathering function.






**Table 1** Determination of erosion and deposition

| Scenario | Condition | Actual erosion $E_a\left(kg\ s^{-1}\right)$ | Deposition $D\left(kg\ s^{-1}\right)$ |
|----------|-----------|-----------------|------------|
| A | $L_{in} + E_p < T_c$ | $T_c - L_{in}$ | 0 |
| B | $L_{in} + E_p \geq T_c$ | $E_p$ | 0 |
| C | $L_{in} \geq T_c$ | 0 | $L_{in} - T_c$ |


**Table 2** Soil grading distribution data used for SSSPAM simulation.

| Grading Range (mm) | | | Ranger1a | Ranger1b |
|------|---|-------|----------|----------|
| 0 | - | 0.063 | 1.40 % | 0.0% |
| 0.063 | - | 0.111 | 2.25 % | 0.0% |
| 0.111 | - | 0.125 | 0.75 % | 0.0% |
| 0.125 | - | 0.187 | 1.15 % | 0.0% |
| 0.187 | - | 0.25 | 1.15 % | 0.0% |
| 0.25 | - | 0.5 | 10.20 % | 0.0% |
| 0.5 | - | 1 | 9.60 % | 0.0% |
| 1 | - | 2 | 12.50 % | 0.0% |
| 2 | - | 4 | 16.40 % | 0.0% |
| 4 | - | 9.5 | 20.00 % | 0.0% |
| 9.5 | - | 19 | 24.60 % | 100.0% |
