# Peer review of "A coupled soilscape-landform evolution model: Model formulation and initial results."

_Earth Surface Dynamics, 2018_

## Referee Comment (RC1) · P.A. Finke (Referee) · 8 May 2018

Journal: ESurf Title: A coupled soilscape-landform evolution model: Model formulation and initial results Author(s): W. D. Dimuth P. Welivitiya et al. MS No.: esurf-2018-16

General The authors describe a quantitative model suitable to estimate evolution of some soil physical properties over the landscape. The model description and the presented mathematical formulations look OK. The manuscript is well-written. My major comments relate to the over-selling of the model as a pedogenesis model (see comment 1), to the linkage to the real world (2, 3) and to the clarity of model assumptions (4).

1. One major comment, even an objection, that I have is that the paper states at many locations that it concerns a soil genesis model. This illustrates a narrow vision on soil genesis, and comes entirely from a geomorphological perspective. In fact, only soil physical processes are considered, and not even all of these (e.g. heat flow and clay migration are no part of the model, the effect of SOC on erodibility is unaccounted for). It ignores that soil genesis involves many other processes, of mineralogical, chemical and biological kinds. See Bockheim and Gennadiyev (2000) for a list of soil formation processes and Minasny et al. (2015; Fig.5) for a check if these processes are covered by the soil models of to date. I therefore advise the authors to be clear in the ambition level of this model, which is the mechanistic simulation of 3D-redistribution of soil particles of various size over the landscape. Mention perhaps "soil texture evolution model", but not soil evolution model s.l.

2. The evolution of the soilscape is only to a limited degree connected to physical boundary conditions such as rain, evaporation, heat/temperature. As I understand it, water plays a role to redistribute topsoil material, but does not influence the subsoil (linkages to weathering of minerals, clay migration). The weathering mechanism entirely concerns physical weathering, and the process is driven by 2 parameters n and alpha, which are empirical (section 2.4). True drivers of physical weathering are related to temperature fluctuations, and specifically the occurrence of frost. For these reasons the model is not fully mechanistic, i.e. does not represent the actual processes, but rather "functional", it describes what happens and uses empirical factors to achieve this. This means that the model cannot be used for studies on effects of global change on soilscapes, where differences in P, PE and T should drive the processes. I would invite the authors to discuss this item in the paper.

3. To allow model testing beyond plausibility testing ("face validity"), which is attempted in the paper, additionally, confrontation to field data would be needed. This is clearly beyond the scope of this paper and, unfortunately, of most soilscape modelling studies. Some sensitivity experiments are done in this paper, which is commendable. I would

expect a strong sensitivity of projected landscapes to the initial landscape as well, but this was not studied. This again touches the ambition level of this model: is it meant for synthetic studies or for real world cases?

4. In general, some assumptions are not so clear. For instance: how does mass redistribution relate to the elevation of the soil-atmosphere interface, in other words, how are mass and volume connected. OK, via the bulk density (for erosion in eq.4; for deposition in eq. 7), but is bulk density then assumed a constant and not affected by bioturbation, strain by weathering? Is this valid over 60.000 years? Are there other assumptions that should be known?

A few specifics:

- l.83: "scorpan" not introduced; this is in fact clorpt+soil point data+position (see McBratney et al. 2003), thus not so different.

- l.573: erosion and d50 correlate: is this a model artefact? For instance, if the organic matter content would be simulated as well, would it not become part of the correlative complex?

- l.689: soil formation and its evolution? =repetition.

Refs:

Bockheim, J.G., Gennadiyev, A.N., 2000. The role of soil-forming processes in the definition of taxa in Soil Taxonomy and the World Soil Reference Base. Geoderma 95, 53–72.

Budiman Minasny, Peter Finke, Uta Stockmann, Tom Vanwalleghem, and Alex McBratney. 2015. Resolving the integral connection between pedogenesis and landscape evolution. Earth-Science Reviews 150: 102-120.

---

## Referee Comment (RC2) · A.J.A.M. Temme (Referee) · 10 May 2018

The manuscript by Welivitiya and co-authors presents a simulation performed with their novel soil-landscape evolution model. The model and the rationale behind it are presented in detail. The model simulation is on a simplified 2D landscape (i.e. a row of cells) representing a plateau over a hillslope and a valley. Two scenarios are simulated, with different depth-dependent weathering functions. Findings are discussed in details, and appear to indicate that the model functions well, and that basic expectations about the joint development of soils and landscapes (co-evolution) are met. The paper is interesting to me as a soil-landscape modeler, and I greatly enjoyed the detailed model

layout and accompanying figures.

I have detailed suggestions in the attached annotations, that amount to minor revisions in and of themselves. Below, I add three general concerns.

1. Although the paper is very interesting to me, I am not sure that it is to the general audience of ESurfD. The meat of the paper is the model presentation, to my mind. That makes me wonder whether an outlet such as Geoscientific Model Development of Computers and Geosciences is not a better choice.

2. The simulation that illustrates the model's performance, is interesting but hypothetical. The paper does a great job of explaining what the results are, but does not give much attention to what that means for how we should think about landscape evolution. This may be possible only in a limited way, given the hypothetical setup, but I do think some comparisons with existing thinking and findings from others are possible. I do one suggestion in the annotations. In another outlet, a lack of connection with existing thinking would be no problem at all, but I think that in ESurfD the readership is particularly interested in that aspect (and perhaps less in model workings).

3. Although the model description is detailed and accurate, it would benefit from pointing out differences with existing models or a few more tradeoffs between accuracy and efficiency, especially where some of the innovative aspects are covered. How do these differ from existing models such as Lorica? I give a few suggestions for improvement.

I am glad to see this work in manuscript form, and I am happy to have it in the public domain so it can be used by colleagues. I wish the authors good luck in considering the changes suggested.

Please also note the supplement to this comment:
https://www.earth-surf-dynam-discuss.net/esurf-2018-16/esurf-2018-16-RC2-supplement.pdf

**ESurfD**

---

## Author Response (AR1)

**General comments for the Editor**

The authors would like to thank the editors of the journal Earth Surface Dynamics for considering publishing our manuscript and for the Referees for their constructive criticism on the manuscript and their valuable suggestions on how to improve the manuscripts overall quality. The responses for each referees comments are presented in separate documents and all comments and suggestions was incorporated into the manuscript where possible. Looking at both the Referees comments the authors felt that we should add a joint response that would alleviate some common concerns from the referees.

The main objective of this manuscript was to introduce the new coupled soilscape-landform evolution model. However we strongly felt that there should be some applications of the model albeit simple, to show how the model performed well in reality and to highlight some of the geomorphic signatures emerging from the modelling results itself. We recognise that it is not a reasonable application that necessarily be directly applicable in the field setting. However we are inspired by the early work on hillslope geomorphology by authors such as *Kirkby* [1971] and *Carson and Kirkby* [1972] which was very useful in understanding hillslope evolution processes. So as first step we used a 1Dimentional hillslope to run our simulations because, understanding dynamics of 1D hillslope evolution is easier and we can better illustrate possible implications for different processes.

In this manuscript emphasis was given to the presenting the model formulation and only limited comparison was done with field data mainly because we were unable to find any experimental work done by any other researcher which has used a similar setup as our simulations. However a subsequent paper will deal with implications of model results in terms of one-dimensional and three-dimensional alluvial fans. In this future manuscript, we compare and contrast the model results with experimental work done by authors such as *Seal et al.* [1997] ,*Toro-Escobar et al.* [2000] and general observations done regarding naturally occurring alluvial fans and their formation dynamics.

We admit that the model is limited in its scientific scope. The model is based on physical fragmentation of parent soil particles and it does not model chemical transformations. The modelling approach used here is complimentary to the chemical weathering modelling work done by Michael J Kirkby [Kirkby, 1977; 1985; 2018]. However we will be incorporating a physically based chemical weathering model described by Willgoose [2018] into SSSAPAM in the future.

At the current time we decided not to consider SOC and its influence in the soil formation and evolution processes. All available evidence suggests that in order to effectively model SOC, it will require an extremely complicated coupled model with soil grading, soil moisture, SOC as well as vegetation. Although formulating such a model is very desirable (and would be an important endeavour by itself) for the entire scientific community, it is well beyond the scope of this current research work.

The deposition model of SSSPAM is designed in such a way that the difference between the
transport capacity and the sediment load of the flow is always deposited regardless of the
settling velocities. This is done to prevent the flow from being over the transport capacity.
Depending on the material grading distribution and the concentration in the profile of the flow,
the theoretical amount of the material that can be deposited can be different. In this model
formulation we assume that the sediment grading is uniform and the sediment concentration is
also uniform within the flow. The reality may not be as simple as that. There is literature that
argues that the sediment concentration profile has an exponential distribution (i.e.most of the
sediment are concentrated near the bottom of the flow) such as *Agrawal et al.* [2012] and that
the grading distribution profile in the flow is also a function of the settling velocity of different
particles (i.e. Larger particles are concentrated near the bottom of the flow).  So in practice the
amount of material deposited at each pixel according to the critical immersion depth might be
higher. Although the approach used in SSSPAM may not perfectly mimic the natural behaviour
of sediment deposition we believe that this is the most effective way to numerically represent
this process in the model at this time.

References.

Agrawal, Y. C., O. A. Mikkelsen, and H. Pottsmith (2012), Grain size distribution and sediment
flux structure in a river profile, measured with a LISST-SL Instrument, *Sequoia Scientific, Inc.*
*Report*.
Carson, M. A., and M. J. Kirkby (1972), Hillslope form and process.

Kirkby, M. (1971), Hillslope process-response models based on the continuity equation, *Inst.*
*Br. Geogr. Spec. Publ*, *3*(1), 5-30.

Kirkby, M. (1977), Soil development models as a component of slope models, *Earth surface*
*processes*, *2*(2-3), 203 -230.

Kirkby, M. (1985), A basis for soil profile modelling in a geomorphic context, *Journal of Soil*
*Science*, *36*(1), 97-121.

Kirkby, M. (2018), A conceptual model for physical and chemical soil profile evolution,
*Geoderma*.

Seal, R., C. Paola, G. Parker, J. B. Southard, and P. R. Wilcock (1997), Experiments on
downstream fining of gravel: I. Narrow-channel runs, *Journal of hydraulic engineering*,
*123*(10), 874-884.

Toro-Escobar, C. M., C. Paola, G. Parker, P. R. Wilcock, and J. B. Southard (2000),
Experiments on downstream fining of gravel. II: Wide and sandy runs, *Journal of Hydraulic*
*Engineering*, *126*(3), 198-208.
Willgoose, G. (2018), *Principles of Soilscape and Landscape Evolution*, Cambridge University
Press.

**Reply for the comments from P.A. Finke (Referee #1) in italic font**

Journal: ESurf Title: A coupled soilscape-landform evolution model: Model formulation and initial results Author(s): W. D. Dimuth P. Welivitiya et al. MS No.: esurf-2018-16

General Comments

The authors describe a quantitative model suitable to estimate evolution of some soil physical properties over the landscape. The model description and the presented mathematical formulations look OK. The manuscript is well-written. My major comments relate to the over-selling of the model as a pedogenesis model (see comment 1), to the linkage to the real world (2, 3) and to the clarity of model assumptions (4).

*General Reply*

*First of all the authors would like to thank the referees for expending their valuable time and energy to review our manuscript. We also greatly appreciate the constructive criticism and the comments of the referees. Referee #1 has raised 4 main issues regarding the current version of the manuscript. The authors will consider all the comments and suggestions made by the referees and accommodate them in the manuscript wherever it is possible. Authors response for each specific comment is presented under each referee comment.*

1. One major comment, even an objection that I have is that the paper states at many locations that it concerns a soil genesis model. This illustrates a narrow vision on soil genesis, and comes entirely from a geomorphological perspective. In fact, only soil physical processes are considered, and not even all of these (e.g. heat flow and clay migration are no part of the model, the effect of SOC on erodibility is unaccounted for). It ignores that soil genesis involves many other processes, of mineralogical, chemical and biological kinds. See Bockheim and Gennadiyev (2000) for a list of soil formation processes and Minasny et al. (2015; Fig.5) for a check if these processes are covered by the soil models of to date. I therefore advise the authors to be clear in the ambi- tion level of this model, which is the mechanistic simulation of 3D-redistribution of soil particles of various size over the landscape. Mention perhaps "soil texture evolution model", but not soil evolution model s.l.

   *The authors do agree with the Referee #1s comment that only limited soil formation processes has been considered in SSSPAM. The model is based on physical fragmentation of parent soil particles and it does not model chemical transformations. The modelling approach used here is complimentary to the chemical weathering modelling work done by Michael J Kirkby [Kirkby, 1977; 1985; 2018]. However we will be incorporating a physically based chemical weathering model described by Willgoose [2018] into SSSAPAM in the future. Also at the current time we decided not to consider SOC and its influence in the soil formation and evolution processes. All available evidence suggests that in order to effectively model SOC, it will require an extremely complicated coupled model with soil grading, soil moisture, SOC as well as vegetation. Although formulating such a model is very desirable for the entire scientific community, it is well beyond the scope of this current research work. Considering the Referee #1s comment and the limited number of soil formation processes simulated in the model we have decided to use "Soil grading evolution" model instead of*

*"pedogenesis" model*

2. The evolution of the soilscape is only to a limited degree connected to physical boundary
conditions such as rain, evaporation, heat/temperature. As I understand it, water plays a
role to redistribute topsoil material, but does not influence the subsoil (linkages to
weathering of minerals, clay migration). The weathering mechanism entirely concerns
physical weathering, and the process is driven by 2 parameters n and alpha, which are
empirical (section 2.4). True drivers of physical weathering are related to temperature
fluctuations, and specifically the occurrence of frost. For these reasons the model is not
fully mechanistic, i.e. does not represent the actual processes, but rather "functional", it
describes what happens and uses empirical factors to achieve this. This means that the
model cannot be used for studies on effects of global change on soilscapes, where
differences in P, PE and T should drive the processes. I would invite the authors to
discuss this item in the paper.

*The authors believe that the Referee #1 may have misunderstood the complexity of the*
*weathering mechanism and how the weathering rate of each soil layer at each pixel*
*(node) is calculated due to the concise manner which it is presented. The 2 parameters*
*mentioned by the Referee #1 only controls the weathering geometry (how many*
*daughter particles and their relative sizes) the weathering rate of each soil layer is*
*controlled by the depth dependent weathering function. The rationale behind these*
*depth dependent weathering functions and how they relate to the "drivers of the*
*physical weathering (specially temperature fluctuations through the soil profile)" are*
*extensively described in Welivitiya et al.[2016]. The authors decided not to repeat the*
*material in previously mentioned paper here due to manuscript length concerns. Also*
*SSSPAM is capable of using depth dependent weathering functions for each simulation*
*node (pixel) depending on the geographical distribution of various physical weathering*
*drivers such as temperature. Also a slight modification to the weathering module in*
*SSSPAM will be able to simulate temporal variations of these weathering drivers as*
*well. So SSSPAM can actually be used to study the effects of global change on soilscapes*
*in the future. For the initial simulations the authors decided to keep the simulation setup*
*as simple as possible to better observe and study how the core parts of the models*
*perform and to see whether the results reflect general trends observed on hillslopes in*
*nature. In later stages more and more capabilities of SSSPAM will be activated and a*
*wide range of multidimensional soilscape simulations will be possible. As the Referee*
*#1 suggests a small paragraph is added to the section to state this fact.*

3. To allow model testing beyond plausibility testing ("face validity"), which is attempted
in the paper, additionally, confrontation to field data would be needed. This is clearly
beyond the scope of this paper and, unfortunately, of most soilscape modelling studies.
Some sensitivity experiments are done in this paper, which is commendable. I would
expect a strong sensitivity of projected landscapes to the initial landscape as well, but
this was not studied. This again touches the ambition level of this model: is it meant for
synthetic studies or for real world cases?

*The authors appreciate the Referee #1s understanding and grasp of the practical*
*difficulties of comparing results produced through a model like SSSPAM with field data.*
*The authors do agree that the simulations presented in this manuscript concerns a*

*hypothetical situation and not much attention was made to compare results with real*
*world scenarios. However the main objective of this manuscript was to introduce the*
*new coupled soilscpae-landform evolution model and demonstrate its ability to co*
*evolve soilscape and landform together resulting in real world trends. So to keep the*
*focus of this manuscript to the model development and model mechanics and to keep*
*the manuscript at a reasonable size, the authors decided not to include a result*
*comparison section to the current manuscript. In fact we have already done some*
*comparison studies of the model simulations (particularly the deposition region of the*
*landforms) with experimental flume scale studies and fluvial fan development and the*
*results will be published in 1 or 2 separate manuscripts in the near future. In these*
*simulations we have identified that the model (even with a reasonable synthetic setup)*
*was able to generate similar trends identified in nature in terms of particle size*
*distribution and landform morphology (particularly for alluvial fans). So we believe*
*that although highly simplified in terms of the number of pedogenetic processes, the*
*model still can be used to explore real world cases.*

4. In general, some assumptions are not so clear. For instance: how does mass
redistribution relate to the elevation of the soil-atmosphere interface, in other words,
how are mass and volume connected. OK, via the bulk density (for erosion in eq.4; for
deposition in eq. 7), but is bulk density then assumed a constant and not affected by
bioturbation, strain by weathering? Is this valid over 60.000 years? Are there other
assumptions that should be known?

*The authors agree that some assumptions made during the model development phase may*
*have been omitted from the manuscript. Revised the manuscript and assumptions are*
*clearly presented*

A few specifics:

- l.83: "scorpan" not introduced; this is in fact clorpt+soil point data+position (see
McBratney et al. 2003), thus not so different.

*- The authors agree that scorpan is in fact a further development of clorpt as the Refree*
*#1 has pointed out. The sentence regarding scorpan was revised and the association of*
*scorpan to clorpt is introduced.*

- l.573: erosion and d50 correlate: is this a model artefact? For instance, if the organic
matter content would be simulated as well, would it not become part of the correlative
complex?
*Yes, all soil components are part of the correlative complex however extensive*
*work has demonstrated that d50 is strongly related to erosion.. However this*
*relationship may be true for natural hillslopes as well due to the process of*
*armoring. i.e High erosion means ability to erode relatively large particles*
*which leads to higher d50. If we incorporate the effects of SOC we would*
*imagine that it will definitely come in to the correlation complex*

-

- l.689: soil formation and its evolution?=repetition.

*There seems to be a repetition in processes describing soil formation and evolution. The*
*sentence was reworded*

- References:

Bockheim, J.G., Gennadiyev, A.N., 2000. The role of soil-forming processes in the
definition of taxa in Soil Taxonomy and the World Soil Reference Base. Geoderma 95,
53–72.

Budiman Minasny, Peter Finke, Uta Stockmann, Tom Vanwalleghem, and Alex McBrat-
ney. 2015. Resolving the integral connection between pedogenesis and landscape
evolution. Earth-Science Reviews 150: 102-12

Kirkby, M. (1977), Soil development models as a component of slope models, Earth
surface processes, 2(2‐3), 203 -230.

Kirkby, M. (1985), A basis for soil profile modelling in a geomorphic context, Journal
of Soil Science, 36(1), 97-121.

Kirkby, M. (2018), A conceptual model for physical and chemical soil profile evolution,
Geoderma.

**Reply for the comments from A.J.A.M. Temme (Referee #2) in italic font**

The manuscript by Welivitiya and co-authors presents a simulation performed with their novel soil-landscape evolution model. The model and the rationale behind it are pre- sented in detail. The model simulation is on a simplified 2D landscape (i.e. a row of cells) representing a plateau over a hillslope and a valley. Two scenarios are simulated, with different depth-dependent weathering functions. Findings are discussed in details, and appear to indicate that the model functions well, and that basic expectations about the joint development of soils and landscapes (co-evolution) are met. The paper is interesting to me as a soil-landscape modeler, and I greatly enjoyed the detailed model layout and accompanying figures.

*General Reply*

*First of all the authors would like to thank the referees for expending their valuable time and energy to review our manuscript. We also greatly appreciate the constructive criticism and the comments of the referees very much. The authors will consider all the comments and suggestions made by the referees and accommodate them in the manuscript wherever it is possible.*

I have detailed suggestions in the attached annotations, that amount to minor revisions in and of themselves. Below, I add three general concerns.

1. Although the paper is very interesting to me, I am not sure that it is to the general audience of ESurfD. The meat of the paper is the model presentation, to my mind. That makes me wonder whether an outlet such as Geoscientific Model Development of Computers and Geosciences is not a better choice.

*The authors do agree that the main objective of this manuscript and the bulk of its content has to do with model formulation and implementation. If the manuscript was just model formulation, we would agree with the Referee #2 s suggestion on submitting this manuscript elsewhere which caters to more mathematically oriented audience. Although a manuscript concerning only model formulation may be interesting, we strongly felt that we needed to include some initial model results (albite simple) to illustrate how the model performed at the simple scale. Also we wanted to highlight some of the geomorphic signatures emerging from the modelling results itself. We believe that publication here is important for not just the modelling community but also for the general geomorphology and soils community who through this work can observe some important physical insights but also raises some important questions. The other important issue is that lack of field data to calibrate/evaluate such models. We hope that we may inspire fieldwork (or reveal existing data) to advance such models as that described here. With this in mind, We strongly believe that Earth Surface Dynamics is the better choice for publishing this manuscript.*

2. The simulation that illustrates the model's performance, is interesting but hypothet- ical. The paper does a great job of explaining what the results are, but does not give much attention to what that means for how we should think about landscape evolution. This may be possible only in a limited way, given the hypothetical setup, but I do think some comparisons with existing thinking and findings from others are possible. I do one suggestion in the annotations. In another outlet, a lack of connection with exist- ing thinking would be no problem at all, but I think that in ESurfD the readership is particularly interested in that aspect (and perhaps less in model workings).

3.

> *The authors do agree that the simulations presented in this manuscript concerns a hypothetical situation and not much attention was made to compare results with real world scenarios. As the Referee #2 has understood the main objective of this manuscript was to introduce the new coupled soilscpae-landform evolution model and demonstrate its ability to co evolve soilscape and landform together resulting in real world trends. So to keep the focus of this manuscript to the model development and model mechanics and to keep the manuscript at a reasonable size, the authors decided not to include a result comparison section to the current manuscript. In fact we have already done some comparison study on the model simulations (particularly the deposition region of the landforms) with experimental flume scale studies and fluvial fan development and the results will be published in 1 or 2 separate manuscripts in the near future.   We also highlight the issue of a lack of field data in the previous comment.*

4. Although the model description is detailed and accurate, it would benefit from pointing out differences with existing models or a few more tradeoffs between accuracy and efficiency, especially where some of the innovative aspects are covered. How do these differ from existing models such as Lorica? I give a few suggestions for improvement.

I am glad to see this work in manuscript form, and I am happy to have it in the public domain so it can be used by colleagues. I wish the authors good luck in considering the changes suggested.

Please also note the supplement to this comment:
https://www.earth-surf-dynam-discuss.net/esurf-2018-16/esurf-2018-16-RC2-supplement.pdf

*Specific comments from the supplemental document*

Referee #2 comment Page 9: I continue to struggle with these issues myself. Propagating resupply seems perfectly reasonable, and is necessary if layer thickness/mass is constant but don't you in this way always provide the top armouring layer with easily erodible soil? I imagine the result is a less effective armour. Does this reflect the real process?

*The authors appreciate the Reviewers comments, understanding and grasp of the practical difficulties of material propagation in a model like SSSPAM. However  according to the method used here the resupplied material to the armour layer is not always easily erodible.*

*Size selectivity only applies to the material movement form the armour layer to the flowing water layer above (i.e. depending on the surface water discharge rate, smaller particles are easily removed from the surface armour layer while larger particles remain). The material resupply to the surface armour occurs from the layer below and the resupplied material have the same grading as the subsurface layer grading (no size selectivity) so both small particles and large particles are resupplied to the armour layer. Most of the time the net effect of this material resupply and the size selective erosion will be enrichment of larger particles and armour strengthening. Depending on the depth dependent weathering function the relative coarseness of the subsurface layers can be less compared to armour layer. But once the armour layer is reconfigured with the added material from below and removal of small particles through erosion, again the net effect is armour strengthening. This layer restructuring is explained later in the paper.*

Referee #2 comment Page 10: If I see it correctly this all means that if you need to deposit 20kg of material and all you have in transport is clays, then you'll deposit some/all of the clays right? Regardless of the settling velocity.

*Yes the referee #2 is correct in recognising this factor. If all there in the transport is clay the amount of material which need to be deposited will be deposited from the available clay. Size selectivity comes in to play when there are a range of particles in the transport (which is almost always the case). Then the larger particles with higher settling velocities will deposit first.*

Referee #2 comment Page 13: Ok, this makes me doubt what I said three pages ago. –at the cost of my understanding. Tell me is $\sum$deposition in a cell equal to $L_{in} - T_c$, which I thought before or is it dictated by the settling velocity? The later seems to allow for $\sum$deposition to be $< L_{in} - T_c$.

*The Referee #2s earlier assumption is correct. The $\sum$deposition at a pixel is always equals to $L_{in} - T_c$. The settling velocities and the critical immersion depths for that matter dictates the distribution of the deposited particles. The Referee 2 is also correct that at the first glance it seems like $\sum$deposition can be $< L_{in} - T_c$. This is why we have the adjustment vector K (equation 5) which ensures*

*$\sum$deposition $= L_{in} - T_c$ (always). The authors previously had a worked example on how this correction vector is calculated. However the authors thought that this may confuse readers with less mathematical/modelling experience and decided not to include it in the final version of the manuscript. The example is again added to the main body of the manuscript. Following is this omitted section with the example.*

 *Calculation of deposition mass vector*

*Following simplified example shows the need to have this adjustment vector and the method*
*we used to calculate it.*

**Table 2** Example calculation of adjustment vector $\underline{K}$.

| Size Class | Elements of $\psi_{in}$ ($\psi_z$) | Entries of $J$ ($J_{z,z}$) | $J_{z,z}\,\psi_z$ | Adjusted $J_{z,z}\,\psi_z$ | Deficit / Surplus | Diagonal elements of $\underline{K}$ | Entries of $\underline{\Phi}$ |
|---|---|---|---|---|---|---|---|
| 1 | 5.00 | 1.0 | 5.00 | 7.29 | -2.29 | -2.29 | 5.00 |
| 2 | 10.00 | 0.7 | 7.00 | 10.21 | -0.21 | -0.21 | 10.00 |
| 3 | 20.00 | 0.4 | 8.00 | 11.67 | 8.33 | 2.00 | 13.67 |
| 4 | 40.00 | 0.1 | 4.00 | 5.83 | 34.17 | 0.50 | 6.33 |
| Total | 75.00 | | 24.00 | 35.00 | | | 35.00 |

*Consider the example values given in Table 2. The total mass of the incoming sediments is 75*
*kg and the sediments are distributed in four size classes. Here the size class one is the largest*
*and has the highest potential for deposition (with $J_{1,1}$ =1) while the size class four has the*
*lowest potential for deposition (with $J_{4,4}$ =0.1). If the transport capacity $T_c$ is 40 kg, 35 kg of*
*incoming sediments should deposit at the pixel as the total deposition D. Using the $\sum J_{z,z}\,\psi_z$*
*value (which is 24) and rescaling these values with D (total deposition mass), we can calculate*
*the masses of sediments which need to be deposited from each grading class. In some cases*
*(when the total deposition D is higher than the $\sum J_{z,z}\,\psi_z$ value) the mass of material which*
*needs to be deposited can be larger than the available sediments in that particular size class.*
*In this example there is 5 kg of sediments in the 1st size class and 10 kg of sediments in the*
*second size class respectively. However, our adjusted calculation dictate that there should be*
*7.29 kg deposition from the 1st size class and 10.21 kg from the 2nd size class which is not*
*possible. So these values needs to be adjusted to reflect maximum possible deposition from size*
*classes one and two which are 5 kg and 10 kg respectively. This adjustment introduces a deficit*
*of 2.5 kg to the total deposition and it needs to be deposited from the 3rd and 4th smaller grading*
*classes. According to the deposition matrix values $J_{z,z}$ the deposition probability ratio between*
*3rd and 4th grading class is 4:1 (0.4:0.1). The deficit mass 2.5kg is deposited from the 3rd and*
*4th size class with 4:1 ratio which accounts to an additional deposition mass of 2 kg from 3rd*
*size class and 0.5 kg from the 4th size class. In this way the entries of the adjustment vector $\underline{K}$*
*are calculated. Depending on the number of size classes and the distribution of the sediments,*
*this adjustment vector $\underline{K}$ needs to be calculated iteratively.*

Referee #2 comment Page 16:  The section 2.3.2 feels excessively detailed and long for a
paper (Better in a model manual perhaps?) consider shortening drastically.

*The authors agree with Referee#2s comment regarding the section 2.3.2. The section is modified and some paragraphs were removed.*

Referee 2 comment Page 17: Also it conveniently avoid possible negative effects of d8. Please comment here, or where you present the d8 choice.

*At the time this manuscript was prepared, the authors had already ran the simulation for 3dimentional synthetic landforms as well. However the authors did not find any significant issues with using d8 for flow calculation.*

Referee #2 comment Page 21: I'm trying to imagine implications. Does it follow from this that nick points cutting back into higher regions slow down less than otherwise expected since they meet fine soft soils as they proceeds further into the plateau? Please comment and harvest for your readers.

*The Referee #2s understanding is correct. When the erosion region cuts back in to the plateau area, weathering process has already broken-down the coarse material in to finer constituents where it can be readily eroded away by the erosion region cutting in to the plateau region. If we did not have weathering the rate at which the erosion region cutting back to the plateau region would be slower as it requires the removal of coarser soil particles. A small discussion mentioning this implication was added to the manuscript as per Referee #2s comment*

Referee #2 comment Page 24: I have the opinion/impression that the discussion section is quite detailed in describing your results. Perhaps a bit much so, and that there is not enough in the way of comparing dynamics with results reported by others, empirically or otherwise.

*As Referee #2s Comments the Discussion section is modified and some paragraphs were removed*

Referee #2 comment Page 26: It would be good to shorten conclusions to about ½ or 2/3 of their size.

*As Referee #2s Comments the conclusions section is modified and some paragraphs were removed*

[revised manuscript text omitted]

*Welivitiya et al.*, 2016b][*W D P Welivitiya et al.*, 2016b]*W D D P Welivitiya et al.* [2016a]
developed a new pedogenesissoil grading evolution model called SSSPAM, which was based
on the approach of mARM3D and showed that the area-slope-$d_{50}$ relationship in *Cohen et al.*
[2009] was robust against changes in process and climate parameters and that the relationship
is also true for all the subsurface soil layers, not just the surface. Although these models predict
the properties of the soil profile at an individual pixel, they do not model the spatial
interconnectivity between different parts of the soil catena resulting from transport-limited
erosion and deposition. Lateral material movement and particle redistribution through deposition is very important in determining the soil characteristics such as soil depth and soil texture [*Chittleborough*, 1992; *Minasny and McBratney*, 2006]. In order to correctly predict  spatially distributed soil attributes and determine the changes in soil attributes with time, coupling soil profile evolution with landform evolution is important.

The first attempt  to integrating soilscape evolution with landform evolution was done by *Minasny and McBratney* [1999; 2001]. They used a single layer to model the influence of soil and weathering processes on landform evolution. In addition to *Minasny and McBratney* [1999; 2001] there are a number of conceptual frameworks found in literature for developing coupled soil profile-landform evolution models [*Sommer et al.*, 2008; *Yoo and Mudd*, 2008]. MILESD [*Vanwalleghem et al.* [2013] is a model which can simulate soil profile evolution coupled with landform evolution. MILESD is built upon the conceptual framework of landscape-scale models for soil redistribution by *Minasny and McBratney* [1999; 2001] and pedon-scale soil formation model developed by *Salvador-Blanes et al.* [2007]. In MILESD the soil profile is divided into four layers containing the bottommost bedrock layer and 3 soil layers above it representing the A, B, and C soil horizons. MILESD was used to model soil development over 60,000 years for a field site in Werrikimbe National Park, Australia [*Vanwalleghem et al.*, 2013]. They matched trends observed in the field such as the spatial variation of soil thickness, soil texture and organic carbon content. A limitation of MILESD is that it only uses three layers to represent the soil profile. Recently the soil evolution module used in MILESD has been modified to incorporate additional layers and has been combined with the landform evolution model LAPSUS to develop a new coupled soilscape-landform evolution model, LORICA [*Temme and Vanwalleghem*, 2015].They found similar results for soil-landform interaction and evolution similar to MILESD simulation results.

Since only three layers were used in MILESD, the representation of the particle size distribution down the soil profile was limited. Although LORICA incorporated additional soil layers into the MILESD modelling framework, detailed exploration of soil profile evolution or interactions between landform evolution and soil profile evolution has not yet been done with this model. Importantly, particle size distribution of the soil can be used as a proxy for various soil attributes such as the soil moisture content [*Arya and Paris*, 1981; *Schaap et al.*, 2001]. The main objective of this paper is to present a new soilscape evolution model capable of predicting the particle size distribution of the entire soil profile by integrating a previously developed soil grading evolution model in to a landscape evolution model.

In previous papers we have presented a pedogenesis model (on a fixed elevation landform) called State Space Soil Production Assessment Model (SSSPAM) [*Welivitiya et al.*, 2016] and explored relationships between the geomorphic parameters slope, contributing area and the soil grading distribution. Similar to previous pedogenesis models such as mARM3D [*Cohen et al.*, 2009; 2010], SSSPAM did not consider the interconnectivity between evolving soil pedons through fluvial processes, no landform evolution was modelled and no changes in the contributing area and slope occurred. In this paper we present the methodology for incorporating sediment transport, deposition and elevation changes of the landform in to SSSPAM modelling framework to create a coupled soilscape-landform evolution model. Detailed information regarding the development and testing of SSSPAM soil grading evolution model is provided in previous papers by the authors ([*Cohen et al.*, 2010; *W D D P Welivitiya et al.*, 2016a][*Sagy Cohen et al.*, 2010; *W D P Welivitiya et al.*, 2016b][*Cohen et al.*, 2010; *W D P Welivitiya et al.*, 2016b][*Cohen et al.*, 2010; *W D P Welivitiya et al.*, 2016b][*
[revised manuscript text omitted]

|---|---|---|---|---|
| 0 | - | 0.063 | 1.40 % | 0.0% |
| 0.063 | - | 0.111 | 2.25 % | 0.0% |
| 0.111 | - | 0.125 | 0.75 % | 0.0% |
| 0.125 | - | 0.187 | 1.15 % | 0.0% |
| 0.187 | - | 0.25 | 1.15 % | 0.0% |
| 0.25 | - | 0.5 | 10.20 % | 0.0% |
| 0.5 | - | 1 | 9.60 % | 0.0% |
| 1 | - | 2 | 12.50 % | 0.0% |
| 2 | - | 4 | 16.40 % | 0.0% |
| 4 | - | 9.5 | 20.00 % | 0.0% |
| 9.5 | - | 19 | 24.60 % | 100.0% |